# SE(3)-HYENA OPERATOR FOR SCALABLE EQUIVARIANT LEARNING

## ABSTRACT

Modeling global geometric context while maintaining equivariance is crucial for accurate predictions in many fields such as biology, chemistry, or when modeling physical systems. Yet, this is challenging due to the computational demands of processing high-dimensional data at scale. Existing approaches such as equivariant self-attention or distance-based message passing, suffer from quadratic complexity with respect to sequence length, while localized methods sacrifice global information. Inspired by the recent success of state-space and long-convolutional models, in this work, we introduce the SE(3)-Hyena operator, the first equivariant network that adopts a long-convolutional framework for geometric systems. SE(3)-Hyena captures global geometric context at sub-quadratic complexity while maintaining equivariance to rotations and translations. Evaluated on the task of all-atom property prediction of large RNA molecules, SE(3)-Hyena matches or outperforms equivariant self-attention while requiring significantly less memory and compute for long geometric sequences. Additionally, we propose equivariant associative recall as a new mechanistic interpretability task for studying the contextual learning capabilities of equivariant models. Notably, our model processes the geometric context of $30k$ tokens $20\times$ faster than the equivariant transformer and allows $72\times$ longer context within the same memory budget. The code will be released upon the acceptance.

## 1 INTRODUCTION

Modeling global geometric context while preserving equivariance is crucial in many real-world tasks. The properties of a protein depend on the global interaction of its residues (Baker & Sali, 2001). Similarly, the global geometry of DNA and RNA dictates their functional properties (Leontis & Westhof, 2001; Sato et al., 2021). In computer vision, modeling global geometric context is crucial when working with point clouds or meshes (Thomas et al., 2018; De Haan et al., 2020). In all these tasks, maintaining equivariance while capturing global context is essential for robust modeling and prediction.

Processing global geometric context with equivariance is challenging due to the computational demands of processing high-dimensional data at scale. Existing methods either rely on global all-to-all operators such as self-attention (Liao & Smidt, 2023; de Haan et al., 2024; Brehmer et al., 2023), which do not scale well due to their quadratic complexity, or they restrict processing to local neighborhoods (Thomas et al., 2018; Köhler et al., 2020; Fuchs et al., 2020), losing valuable global information. This limitation is a significant practical bottleneck, necessitating more efficient solutions for scalable equivariant modeling with a global geometric context.

An efficient algorithm for modeling global context should support parallelization during training while maintaining bounded computational costs relative to sequence length during inference. One approach involves recurrent operators (Orvieto et al., 2023; De et al., 2024), which provide bounded compute but lack easy parallelization. Another family of methods relies on self-attention (Vaswani et al., 2017) allowing parallel processing at the cost of quadratic computational complexity. The most recent advances leverage state-space (Gu et al., 2021b; Fu et al., 2022; Gu & Dao, 2023) and long-convolutional (Romero et al., 2021; Poli et al., 2023) frameworks, enabling global context reasoning in sub-quadratic time with easy parallelization. Extending these models to accommodate equivariance remains an unexplored direction.

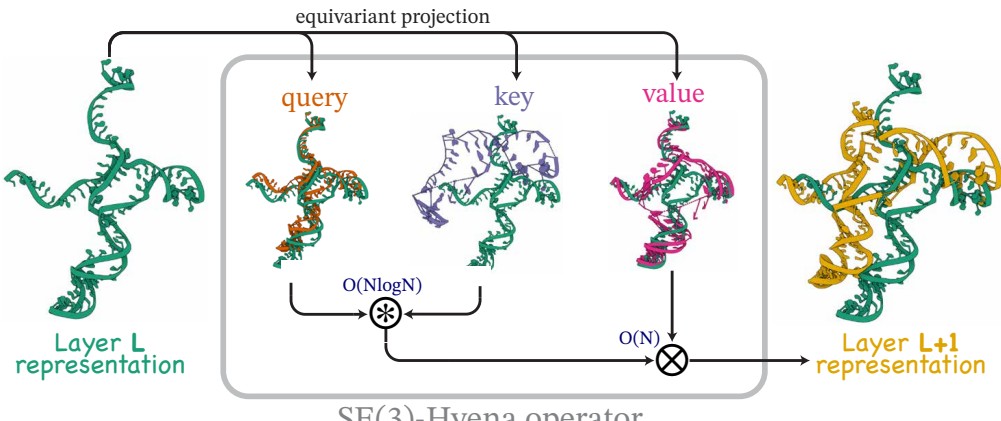

Figure 1: **Information flow in the SE(3)-Hyena operator.** The learned vector **query**, **key**, **value** tokens for a large-molecule atomic system from Rhiju Das (2020). Vector query, key, and value capture various geometric facets of the molecule and interact via vector long convolution and element-wise product (denoted as ⊛ and ⊗ respectively in the Figure) . The resulting operator is scalable due to $O(N \log N)$ computational complexity of the long convolution. All structures are plotted on top of the Layer L representation structure for comparison.

Inspired by the success of state-space and long-convolutional methods, in this work, we propose an SE(3)-Hyena operator that efficiently models global geometric context in sub-quadratic time while preserving equivariance to rotations and translations. The core of our method is an equivariant vector long convolution that utilizes cross products between equivariant queries and keys, as shown in Figure 1. The vector long convolution is implemented in the Fourier domain, achieving $O(N \log N)$ computational complexity. We then demonstrate how to further extend the vector long convolution to geometric long convolution by accommodating interaction between invariant and equivariant subspaces to extract more complex geometric relations. The SE(3)-Hyena operator integrates the geometric long convolution with equivariant projection layer, selective gating, and key-value normalization that, as we demonstrate, is essential for stability.

We evaluate SE(3)-Hyena against other local and global equivariant methods on the tasks of dynamical system modeling and all-atom large molecule property prediction. In addition, we test the proposed model on the equivariant associative recall task, a geometric counterpart of mechanistic interpretability inductive heads problem (Olsson et al., 2022) that we introduce to study contextual learning capabilities of equivariant models. Our results suggest that SE(3)-Hyena outperforms local methods and performs on par with other global methods while requiring significantly less memory and compute for long sequences. In particular, for a sequence of $30k$ tokens, the equivariant Hyena runs $20\times$ faster than equivariant self-attention methods. Notably, when the equivariant transformer runs out of memory on sequences over $37k$ tokens, *our model can handle up to $2.7M$ million tokens on a single GPU*, providing up to $72\times$ longer context length within the same computational budget.

To sum up, we make the following contributions:

- We propose the SE(3)-equivariant Hyena operator which enables modeling global geometric context in sub-quadratic time. Through the ablation study, we identify several key insights necessary to scale the model for large-scale problems.

- We demonstrate that SE(3)-Hyena outperforms local equivariant methods and matches or outperforms equivariant self-attention on several large-scale molecular property prediction and dynamical system modeling tasks while requiring significantly less memory and compute for long sequences.

- We propose a novel equivariant associative recall task as a geometric variant of the mechanistic interpretability of equivariant models.

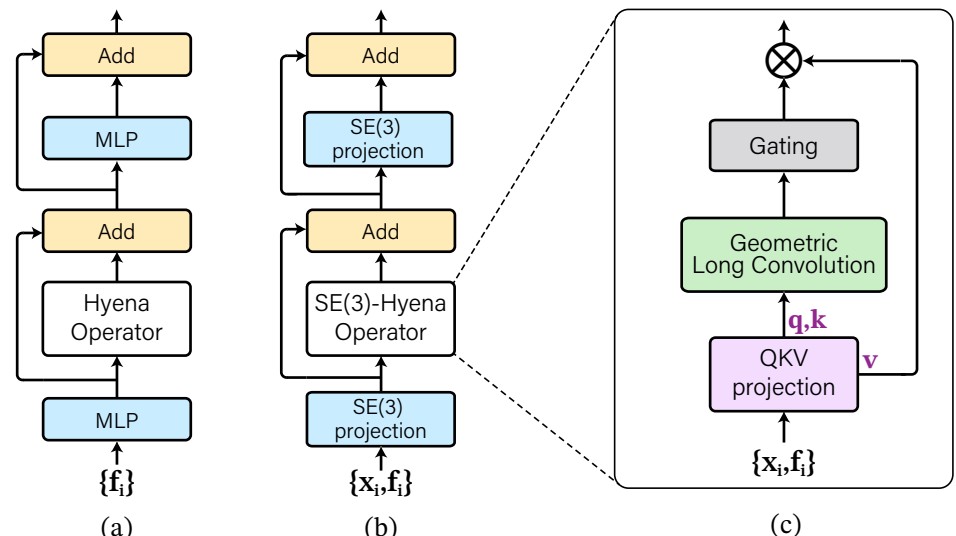

(a)    (b)    (c)

Figure 2: **SE(3)-Hyena building blocks.** **(a)** Schematic of existing Hyena architecture Poli et al. (2023). **(b)** The proposed architecture consists of the SE(3)-Hyena operator. **(c)** The architecture of the SE(3)-Hyena operator includes query, key, value projection, gating, and geometric long convolution for global context aggregation.

## 2  SE(3)-HYENA OPERATOR

The SE(3)-Hyena operator is designed for tasks that require modeling invariant and equivariant features in geometric graphs. A geometric graph of $N$ nodes is represented by a set of features $\{\mathbf{x}_i, \mathbf{f}_i\}_{i=1}^N$ where $\mathbf{x}_i \in \mathbb{R}^3$ represents vector features (e.g. coordinates or velocities), and $\mathbf{f}_i \in \mathbb{R}^d$ represents scalar features (e.g. atom types, charges or fingerprints). We call $\mathbf{x}_i$ *geometric or vector token*, and $\mathbf{f}_i$ is a *scalar token*. In addition, a geometric graph can have edge attributes $e_{ij} \in \mathbb{R}^e$ between two nodes (e.g. relative distance) which are treated as scalar features. When working with geometric graphs, a neural network must respect symmetries of the input space and hence be equivariant with respect to geometric tokens and invariant with respect to scalar tokens.

The SE(3)-Hyena operator consists of invariant and equivariant streams responsible for processing scalar and vector tokens respectively. Formally, the SE(3)-Hyena operator $\Psi : \mathbb{R}^3 \times \mathbb{R}^d \to \mathbb{R}^3 \times \mathbb{R}^d$ satisfies equivariance property:

$$\{L_g(\hat{\mathbf{x}}_i), \hat{\mathbf{f}}_i\}_{i=1}^N = \Psi\left(\{L_g(\mathbf{x}_i), \mathbf{f}_i\}_{i=1}^N\right) \tag{1}$$

where $L_g : \mathbb{R}^3 \to \mathbb{R}^3$ is a representation of a group action $g \in SE(3)$. Thus, geometric tokens $\mathbf{x}_i$ transform accordingly with the group action while scalar tokens $\mathbf{f}_i$ remain invariant.

We make the overall information flow similar to the information flow of a transformer, as illustrated in Figure 2. The input is first mapped into keys, queries, and values via the projection layer. Next, the global context is aggregated via long convolution. Finally, the result of the long convolution is processed by the selective gating mechanism, and multiplied with values. At the end, the residual connection is added and the result is passed through the output projection layer. We design each layer to be equivariant with respect to transformations of geometric tokens, and invariant for scalar tokens. This way, a model consisting of a composition of equivariant layers is equivariant (Weiler & Cesa, 2019). We provide a formal proof of equivariance for each SE(3)-Hyena module in the Appendix 6.5.2.

### 2.1  EQUIVARIANT PROJECTION LAYER

The equivariant projection layer $\phi : \mathbb{R}^3 \times \mathbb{R}^d \to \mathbb{R}^3 \times \mathbb{R}^d$ is a key component of the SE(3)-Hyena operator. It serves three primary functions: embedding scalar and vector tokens, mapping embedded tokens into query, key, and value triplets and serving as an output projection layer. Because we

have scalar and vector features, the projection layer should embed both of them into hidden scalar and vector features while maintaining equivariance. Importantly, our framework is flexible and can accommodate *any equivariant network as the projection function*. We discuss our specific choice of projection function in Section 3.

**Projecting into queries, keys, and values**   Similar to transformers, we project tokens into queries, keys, and value triples. Because our SE(3)-Hyena operator consists of equivariant and invariant streams, we need to obtain queries, keys, and values for invariant and equivariant features. To this end, we utilize the equivariant projection layer to emit scalar and vector queries $\mathbf{q}_i^{inv} \in \mathbb{R}^d$, $\mathbf{q}_i^{eqv} \in \mathbb{R}^3$ as $\mathbf{q}_i^{inv}, \mathbf{q}_i^{eqv} = \phi_Q(\mathbf{x}_i, \mathbf{f}_i)$ for $i$-th token, and similarly key and value projection layers $\phi_K, \phi_V$ for scalar and vector keys $\mathbf{k}_i^{inv}, \mathbf{k}_i^{eqv}$ and values $\mathbf{v}_i^{inv}, \mathbf{v}_i^{eqv}$ respectively.

## 2.2 Geometric long convolution

Geometric long convolution serves as the global context aggregation module in the SE(3)-Hyena operator, analogous to the self-attention mechanism in transformer architectures. Its primary function is to encode global relations between query and key tokens. The geometric long convolution comprises two key components: scalar long convolution that accommodates global relations between scalar invariant tokens and equivariant vector long convolution that handles global relations between vector tokens. The scalar and vector convolutions can be used as is for global context aggregation, or as we demonstrate, they can be combined into the geometric long convolution enabling the interaction between invariant and equivariant subspaces.

**Scalar long convolution**   For global context aggregation of invariant scalar features, we employ long convolution (Romero et al., 2021; Poli et al., 2023) between query and key tokens. Queries serve as the input signal projection, while keys form a data-controlled implicit filter. To reduce computational complexity, we use circular FFT-convolution, similar to Romero et al. (2021); Poli et al. (2023). Let $\mathbf{q}^{inv}$ and $\mathbf{k}^{inv}$ be two sequences of length $N$ composed of sets of one-dimensional invariant queries $\{q_i^{inv}\}_{i=1}^N$ and keys $\{k_i^{inv}\}_{i=1}^N$ respectively. Then, the global context for scalar tokens is aggregated by the FFT-convolution as:

$$\mathbf{u}^{inv} = \mathbf{q}^{inv} \circledast \mathbf{k}^{inv} = \mathbf{F}^H \text{diag}(\mathbf{F}\mathbf{k}^{inv})\mathbf{F}\mathbf{q}^{inv} \tag{2}$$

where $\mathbf{F}$ is a discrete Fourier transform matrix, and $\text{diag}(\mathbf{F}\mathbf{k}^{inv})$ is a diagonal matrix containing Fourier transform of the implicit filter $\mathbf{k}^{inv}$. In the case when query's and key's dimension $d > 1$, the scalar FFT-convolution runs separately for each dimension, rendering computational complexity of $O(dN \log N)$ sub-quadratic in sequence length.

**Vector long convolution**   To enable global context aggregation for geometric tokens, we introduce an equivariant vector long convolution. Unlike scalar convolutions that use dot products, vector convolution operates with vector cross product, denoted as $\times$, between vector signals. Given a vector signal consisting of $N$ vector tokens $\mathbf{q}^{eqv} \in \mathbb{R}^{N \times 3}$ and a vector kernel $\mathbf{k}^{eqv} \in \mathbb{R}^{N \times 3}$, we define the vector long-convolution as:

$$\mathbf{u}_i^{eqv} = (\mathbf{q}^{eqv} \circledast_\times \mathbf{k}^{eqv})_i = \sum_{j=1}^N \mathbf{q}_i^{eqv} \times \mathbf{k}_{j-i}^{eqv} \tag{3}$$

A naive implementation of convolution is Equation 3 has quadratic complexity. However, we show how it can be formulated as a series of scalar convolutions that can be efficiently carried out by the FFT. This is due to the fact that a cross product can be written element-wise through the series of scalar products as $(\mathbf{a} \times \mathbf{b})[l] = \varepsilon_{lhp}\mathbf{a}[h]\mathbf{b}[p]$ where $\varepsilon$ is *Levi-Civita* symbol, and $\mathbf{a}[h]$ denotes a projection onto $h$-th basis vector. Consequently, the $l$-th component of the vector convolution in Equation 3 can be written element-wise as:

$$(\mathbf{q}^{eqv} \circledast_\times \mathbf{k}^{eqv})_i [l] = \varepsilon_{lhp} \sum_{j=1}^N \mathbf{q}_i^{eqv}[h] \, \mathbf{k}_{j-i}^{eqv}[p] = \varepsilon_{lhp} \left(\mathbf{q}^{eqv}[h] \circledast \mathbf{k}^{eqv}[p]\right)_i \tag{4}$$

Thus, we can obtain $l$-th component of a resulting vector signal via a scalar convolution over the $h$-th and $p$-th components of the sequences $\mathbf{q}^{eqv}$ and $\mathbf{k}^{eqv}$ respectively. Since the scalar convolution can be implemented with the FFT, decomposing the vector convolution to the series of scalar convolutions allows reducing its quadratic complexity to $O(N \log N)$.

**Invariant-Equivariant subspace interaction** Scalar and vector long convolutions aggregate global context for scalar and vector tokens separately, limiting the complexity of global relations that can be modeled. To address this limitation, we introduce geometric long convolution, which combines scalar and vector convolutions to enable scalar-vector interactions. Our approach expands possible interactions between scalar-vector tuples $(\alpha_1, \mathbf{r}_1)$ and $(\alpha_2, \mathbf{r}_2) \in \mathbb{R} \times \mathbb{R}^3$ beyond basic scalar $(\alpha_1 \alpha_2)$ and vector cross $(\mathbf{r}_1 \times \mathbf{r}_2)$ products. In addition, we incorporate scalar-vector products $(\alpha_1 \mathbf{r}_2, \alpha_2 \mathbf{r}_1)$ and vector dot products $(\mathbf{r}_1^T \mathbf{r}_2)$ to represent invariant-equivariant token interactions. With this, we define the mapping from input tokens to resulting scalar and vector tokens as $\alpha_3 = \lambda_1 \alpha_1 \alpha_2 + \lambda_2 \mathbf{r}_1^T \mathbf{r}_2$ and $\mathbf{r}_3 = \lambda_3 \alpha_1 \mathbf{r}_2 + \lambda_4 \alpha_1 \mathbf{r}_2 + \lambda_5 (\mathbf{r}_1 \times \mathbf{r}_2)$ where $\lambda_i$ are trainable weights to learn the contribution of each product factor. Figure 3 illustrates this information flow between input and output tokens. Our approach is reminiscent of a convolution operation based on the Clebsch-Gordan tensor product between two [type-0, type-1] high-order tensors (Rowe & Bahri, 2000). However, we maintain tokens in Cartesian space to avoid the high computational complexity associated with spherical representations. We derive the generalization to higher-order steerable tensors in Appendix 6.6.

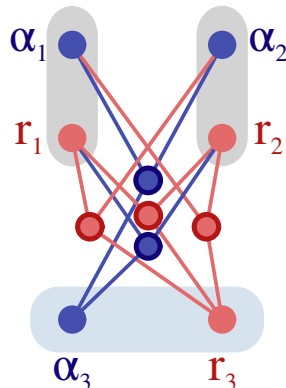

Figure 3: Scalar-vector interactions in geometric long convolution. **Blue** lines represent interactions leading to a scalar output $\alpha_3$, and **red** lines are interactions leading to a vector output $\mathbf{r}_3$.

To implement geometric long convolution efficiently, we first map the $d$-dimensional invariant tokens $\mathbf{f}_i$ to scalars using a linear layer: $\tilde{f}_i = \mathbf{w}^T \mathbf{f}_i$ with $\mathbf{w} \in \mathbb{R}^d$. This results in feature tuples $(\tilde{f}_i, \mathbf{x}_i) \in \mathbb{R} \times \mathbb{R}^3$ for $i$-th input token. Next, note that there are 2 interactions contributing to the scalar output and 3 interactions leading to vector output. Since those interactions only involve either scalar or cross products, and we know how to perform long convolution with those operations as shown earlier, we can obtain the output of the geometric long convolution by leveraging scalar and vector long convolutions and, as a result, maintaining a sub-quadratic complexity. We provide more details on the specific implementation of the geometric long convolution in Appendix 6.5.1.

## 2.3 SELECTIVE GATING

The SE(3)-Hyena model incorporates a selective gating mechanism to dynamically control information flow, similar to the softmax operation in self-attention. This mechanism enables the model to emphasize relevant tokens while suppressing less informative ones. The gating is implemented through the gating function $\gamma : \mathbb{R}^3 \times \mathbb{R}^d \to [0, 1]$ that predicts soft masking value between 0 and 1 for each token. Practically, we implement the gating employing a projection layer (Section 2.1) followed by a linear layer $\mathbb{R}^d \to \mathbb{R}$ with sigmoid activation. The gating function emits a masking value $m_i$ for $i$-th token in a sequence. The masking is applied to the output of the geometric long convolution as $m_i \mathbf{u}_i^{inv}$ and $m_i \mathbf{u}_i^{eqv}$ for scalar and vector tokens.

Finally, the resulting gated tokens are then integrated with value tokens $\mathbf{v}_i^{inv}$ and $\mathbf{v}_i^{eqv}$ using the element-wise product for scalar tokens and cross product for vector tokens. The SE(3)-Hyena operator is thus composed of input projection, QKV-projection, geometric long convolution, and gating. We provide the ablation study on SE(3)-Hyena operations in Appendix 6.4.

## 3 REFINING SE(3)-HYENA OPERATOR IMPLEMENTATION

**Equivariant projection with local and global context** Inspired by the insights from planar graph data processing (Rampášek et al., 2022) where alternating global and local context processing significantly improves convergence, we found that the same holds also for geometric graphs. For projection

with context, we employ a one-layer Equivariant Graph Neural Network (EGNN) (Satorras et al., 2021). This method maintains equivariance, allows interaction between invariant and equivariant sub-spaces, incorporates local contex and easily accommodates edge features when available. We further enhance the EGNN by including global context tokens that summarize the entire geometric graph. This is in line with the recent developments in planar GNNs, where the approach of using virtual nodes representing global context has shown to provide significantly better convergence (Southern et al., 2024; Hwang et al., 2022).

Given $N$ input tokens $\{\mathbf{x}_i, \mathbf{f}_i\}_{i=1}^N$, $G$ global context tokens $\{\mathbf{g}_i, \mathbf{h}_i\}_{i=1}^G$ where $\mathbf{g}_i \in \mathbb{R}^3$ and $\mathbf{h}_i \in \mathbb{R}^d$, the equivariant projection layer follows the message passing framework:

$$
\begin{array}{cc}
\text{Local message} & \text{Global message} \\
\mathbf{m}_{ij}^{loc} = \varphi_l(\mathbf{f}_i, \mathbf{f}_j, \|\mathbf{x}_i - \mathbf{x}_j\|_2, e_{ij}) & \mathbf{m}_{ij}^{glob} = \varphi_g\big(\mathbf{f}_i, \mathbf{h}_j, \log(1 + \|\mathbf{x}_i - \mathbf{g}_j\|_2)\big) \\
\mathbf{m}_i^{loc} = \sum_{j \in \mathcal{N}_r^k(i)} \mathbf{m}_{ij}^{loc} & \mathbf{m}_i^{glob} = \sum_{j=1}^G \mathbf{m}_{ij}^{glob} \\
\\
\text{Vector update} & \text{Scalar update} \\
\hat{\mathbf{x}}_i = \mathbf{x}_i + \frac{1}{|\mathcal{N}_r^k(i)|} \sum_{j \in \mathcal{N}_r^k(i)} (\mathbf{x}_i - \mathbf{x}_j)\varphi_x(\mathbf{m}_{ij}^{loc}) & \hat{\mathbf{f}}_i = \varphi_f\big(\mathbf{f}_i, \mathbf{m}_i^{loc} + \mathbf{m}_i^{glob}\big)
\end{array}
$$

where $\mathcal{N}_r^k(i)$ represents the top-k neighbors of the i-th token within radius $r$. The message functions $\varphi_g, \varphi_l : \mathbb{R}^d \times \mathbb{R}^d \times \mathbb{R}^1 \times \mathbb{R}^e \to \mathbb{R}^d$ and feature update functions $\varphi_f : \mathbb{R}^d \times \mathbb{R}^d \to \mathbb{R}^d$ and $\varphi_x : \mathbb{R}^d \to \mathbb{R}^1$ are parameterized as one-layer perceptrons. We use log-distances for global messages to enhance stability, addressing large absolute distances that can occur with global tokens. For simplicity, we compute messages only between local tokens and from global-to-local tokens, omitting local-to-global messages and updates to global tokens.

**Global context tokens**   Global context tokens provide information about the overall structure of geometric graphs. While representing a compressed snapshot, they make scalar and vector feature updates aware of their global position. We create this snapshot by taking a weighted average of scalar or vector tokens across all input tokens. Given input tokens $\{\mathbf{x}_i, \mathbf{f}_i\}_{i=1}^N$, a set of $G$ global tokens $\{\mathbf{g}_j, \mathbf{h}_j\}_{j=1}^G$ is obtained as $\mathbf{g}_j = C_j^{-1} \sum_{i=1}^N \omega_{ij}\mathbf{x}_i$ for vector tokens and $\mathbf{h}_j = C_j^{-1} \sum_{i=1}^N \omega_{ij}\mathbf{f}_i$ for scalar tokens where $C_j = \sum_{i=1}^N \omega_{ij}$. The weighting factor $\omega_{ij}$ is a learned scalar representing the contribution of the $i$-th input token to the $j$-th global context token. To detach the learning process from specific sequence length requirements, we employ a small SIREN network (Sitzmann et al., 2020) to predict $\omega_{ij}$ for each position.

**Key-Value normalization**   Key-Value normalization is crucial for maintaining numerical stability in the SE(3)-Hyena operator. To understand its necessity, let $M$ be the maximum magnitude of an element in the query $\mathbf{q}$, key $\mathbf{k}$, or value $\mathbf{v}$ tokens. With this, the output magnitude after the long convolution can be bounded as $\|\mathbf{u}\|_2 \leq \|\mathbf{q}\|_2\|\mathbf{k}\|_2 \leq M^2$. When further multiplied with the values, the final output magnitude is bounded as $\|\mathbf{y}\|_2 \leq \|\mathbf{u}\|_2\|\mathbf{v}\|_2 \leq M^3$. This cubic growth with respect to input magnitudes leads to numerical instability and data type overflows in the SE(3)-Hyena operator. Consequently, training with this naive implementation requires extremely small learning rates and gradient clipping, resulting in very slow convergence. To address this issue, we employ key-value normalization, ensuring unit norm for keys and values: $\|\mathbf{k}\|_2 = \|\mathbf{v}\|_2 = 1$. This bounds the output magnitude as $\|\mathbf{y}\|_2 \leq \|\mathbf{q}\|_2 \leq M$ removing the unstable cubic bound. In practice, we observed using key-value normalization to be a critical step to ensure convergence and numerical stability of the SE(3)-Hyena operator.

## 4 EXPERIMENTS

### 4.1 EQUIVARIANT ASSOCIATIVE RECALL

Associative recall is one of the standard mechanistic interpretability tasks used for quantifying the contextual learning capabilities of sequence models Olsson et al. (2022). In this task, a model is required to perform associative recall and copying; for instance, if a model previously encounters

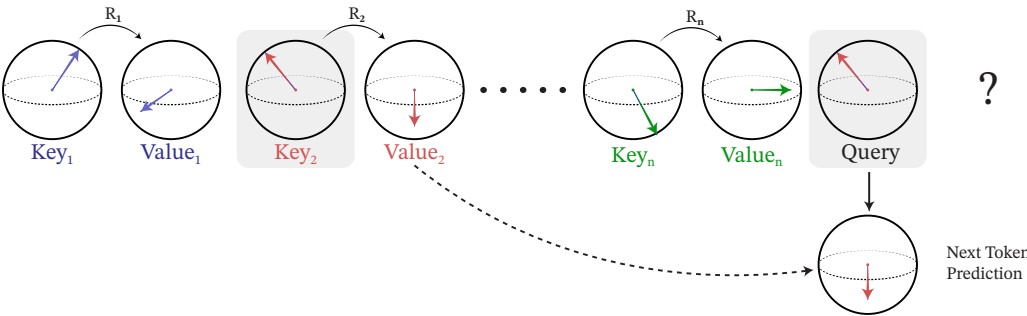

Figure 4: **Equivariant associative recall task.** An equivariant associative recall requires retrieving a vector token for a given vector query based on the context. The retrieval mechanism requires equivariance to rotation of tokens in a sequence. As standard associative recall serves to test the capability of models to learn global context, the equivariant associative recall task serves to test capability of models to learn global context with equivariance.

the bigram "Harry Potter" in a sequence, it should accurately predict "Potter" the next time "Harry" appears, by referencing its past occurrence Gu & Dao (2023).

To adapt this for geometric contexts, we modify the standard associative recall to accommodate 3D vectors. In this version, the tokens within a bigram (key and value) relate to each other by a rotation matrix. The model processes a sequence of $N$ vector tokens concluding with a query token and must predict a vector associated with this query seen earlier in the sequence, as illustrated in Figure 4. The task ensures that rotating the entire sequence affects only the orientation of the predicted vector, not the key-value relationship within a bigram, thereby making the task equivariant to rotations. The task complexity depends on the number of tokens in a sequence and on vocabulary size where the vocabulary items are unique key-value vector bigrams.

We propose two versions of the equivariant associative recall task: random and fixed vocabulary versions. In the random vocabulary version, sequences are sampled from a randomly generated vocabulary at each training iteration while validation and test sets are fixed. Thus, an only way for a model to solve this task is to learn an equivariant retrieval mechanism that can associate a given query with a corresponding value vector. In the fixed vocabulary version, the vocabulary is fixed and shared among training, test and validation sets, and the sequences are randomly rotated during the training. This way, the fixed version tests the model's ability to learn underlying vocabulary and generalize it for various orientations.

**Implementation details**    We evaluate models on sequences of 3D vector tokens. Sequences lengths vary from $2^7$ to $2^{10}$ vector tokens. We use 2600 sequences for training and 200 sequences for each test and validation. We compare equivariant models that permit global context: Vector Neuron Transformer VNT (Deng et al., 2021; Assaad et al., 2022), Equiformer (Liao & Smidt, 2023), SE(3)-transformer (Appendix 6.7) and proposed SE(3)-Hyena. We use 3 layers for all models. We use hidden dimensions of 80 for SE(3)-Hyena, and for other methods, the hidden dimension is set to equate a number of parameters across the models. All models are trained for 400 epochs using Adam optimizer. Detailed setup is in Appendix 6.3. Note that we run Equiformer only on sequence lengths $2^7$ and $2^8$ as it runs out of memory on longer contexts.

**Results**    The results for various sequence lengths are presented in Figure 5. We record the mean squared error between predicted and ground truth vectors as a performance measure. For a fixed vocabulary variant, all equivariant models perform on par across the whole range of evaluated sequence lengths. This demonstrates that in this setting, all tested equivariant global context mechanisms can learn an underlying vocabulary and can generalize it for various orientations. At the same time, we observed that non-equivariant baselines can only learn the expectation across the training dataset.

For the random vocabulary variant, we observed that SE(3)-Hyena and SE(3)-Transformer can successfully learn the retrieval function to associate a target query with a corresponding target value vector, although the error is higher than in the fixed vocabulary setting. Interestingly, we observed that both the VNT and Equiformer models struggled to learn the retrieval function, performing

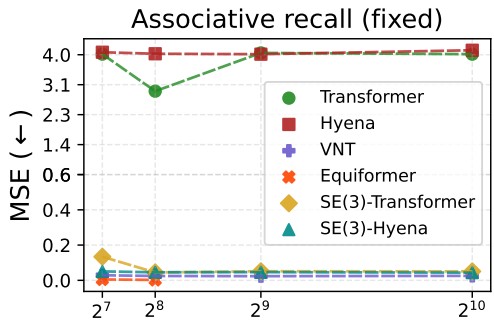 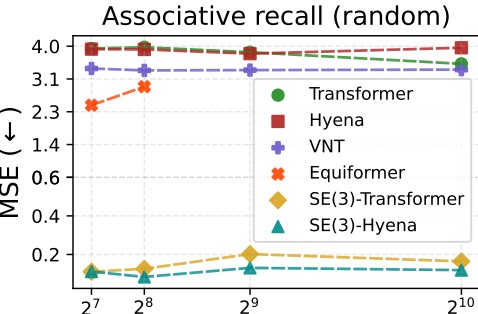

Figure 5: **Left:** The MSE between retrieved and target vectors for the fixed vocabulary associative recall task is plotted across various sequence lengths. Equivariant models effectively learn and generalize the underlying vocabulary across different orientations. **Right:** The MSE for the random vocabulary associative recall task. The SE(3)-Hyena excels in learning the equivariant retrieval function, successfully associating target queries with their corresponding value vectors.

only slightly better than non-equivariant baselines. We attribute the superior performance of the SE(3)-Hyena in this task to the alternating usage of both local and global context, so the model can first associate key-value bi-grams locally, and then compare those bi-grams on a whole sequence level. This is further confirmed by the strong performance of the SE(3)-Transformer model where the only change relative to SE(3)-Hyena is the usage of self-attention operation instead of the long convolution. Also, similarly to the fixed vocabulary setting, we observed that non-equivariant models could only learn the expectation vector across the training dataset. For both fixed and random vocabulary settings, we observed faster converges for longer sequences which can be attributed to a higher frequency of target key-value bigram occurrence which potentially allows more training examples for global context models.

## 4.2 All-atom RNA property prediction

We evaluate SE(3)-Hyena on large molecule molecule property prediction, focusing on RNA molecules that come with rotational and translational symmetries of atom coordinates. This application is crucial given RNA's rapidly growing therapeutic importance, where accurate in-silico modeling can significantly reduce costly and time-consuming in-vitro/in-vivo measurements. RNA molecules present unique challenges: their high dimensionality (hundreds of nucleotides, thousands of atoms) and the lack of established backbone extraction methods often necessitate processing large-scale all-atom systems. In addition, RNA molecules form complex secondary and tertiary structures, with properties influenced by both short- and long-range structural interactions. Consequently, an effective model must combine equivariance, multi-scale context processing, and scalability to handle these intricate, high-dimensional structures.

**Datasets**  We use two real-world RNA datasets: Open Vaccine Covid-19 (Rhiju Das, 2020) and Ribonanza-2k (He et al., 2024). Open Vaccine (3000 sequences) provides stability profiles (reactivity, degradation at `pH10`, and degradation at `Mg pH10`) at various nucleotides, and contains sequences up to 7800 nodes in all-atom resolution. Ribonanza-2k (2260 sequences) offers reactivity profiles for `DMS` and `2A3` chemical modifiers (Low & Weeks, 2010). These profiles are crucial for understanding RNA behavior in different environments and essential for RNA-based therapeutics design. The Ribinanza-2k dataset contains mRNA sequences of up to 11300 atoms. We generate 3D structures using RhoFold (Shen et al., 2022), and we focus on all-atom and on eta-theta pseudotorsional backbone (Wadley et al., 2007) representations. Detailed dataset preparation is in Appendix 6.3.

**Implementation details**  We use atom and nucleotide identities as invariant features and 3D coordinates as equivariant features. The models process all-atom representations of RNA, outputting stability and degradation profiles for each nucleotide. We compare our SE(3)-Hyena against other models with a local or global context. As local context baselines, we employ standard SchNet, TFN, EGNN, and more recent Torch-MD Equivariant Transformer (TMD-ET) (Pelaez et al., 2024; Thölke & Fabritiis, 2022), LEFTNet (Du et al., 2023) and FastEGNN (Zhang et al., 2024). As global context baselines, we employ Vector Neuron Transformer (VNT) (Assaad et al., 2022), Equiformer (Liao & Smidt, 2023) and the SE(3)-Transformer model described in Appendix 6.7. Models are trained in a

Table 1: Comparison of RNA large molecule property prediction methods across Open Vaccine Covid-19 and Ribonanza datasets. Local context methods are highlighted in red, and global context methods are highlighted in cyan. Values represent mean RMSE ± standard deviation (lower is better). Best results in each category are in **bold**.

| Method | Open Vaccine Covid-19 | | | | Ribonanza-2k | | |
|---|---|---|---|---|---|---|---|
| | React. | Deg. (pH10) | Deg. (Mg pH10) | Avg. | React. (DMS) | React. (2A3) | Avg. |
| *Backbone representation* | | | | | | | |
| SchNet | 0.427±0.003 | 0.604±0.007 | 0.473±0.007 | 0.501±0.008 | 0.850±0.014 | 0.886±0.006 | 0.868±0.006 |
| TFN | 0.421±0.001 | 0.612±0.006 | 0.498±0.005 | 0.510±0.006 | 0.857±0.005 | 0.823±0.014 | 0.840±0.006 |
| EGNN | 0.420±0.003 | 0.598±0.007 | 0.468±0.007 | 0.495±0.008 | 0.778±0.014 | 0.833±0.002 | 0.805±0.011 |
| FastEGNN | 0.422±0.003 | 0.610±0.007 | 0.492±0.007 | 0.508±0.009 | 0.860±0.014 | 0.918±0.016 | 0.889±0.030 |
| LEFTNet | 0.424±0.003 | 0.612±0.007 | 0.488±0.007 | 0.508±0.009 | 0.850±0.014 | 0.904±0.006 | 0.877±0.002 |
| TMD-ET | 0.419±0.002 | 0.594±0.006 | 0.487±0.006 | 0.500±0.006 | 0.774±0.006 | 0.788±0.006 | 0.781±0.006 |
| SE(3)-Trans. | 0.323±0.004 | 0.442±0.008 | 0.445±0.005 | 0.403±0.006 | 0.700±0.013 | 0.614±0.009 | 0.657±0.011 |
| VNT | 0.320±0.010 | 0.443±0.010 | 0.440±0.010 | 0.401±0.008 | 0.695±0.010 | 0.611±0.010 | 0.653±0.004 |
| Equiformer | 0.322±0.010 | 0.446±0.010 | 0.459±0.010 | 0.409±0.008 | 0.689±0.010 | 0.609±0.010 | 0.649±0.004 |
| **SE(3)-Hyena** | **0.307**±0.043 | **0.436**±0.047 | **0.345**±0.044 | **0.363**±0.045 | **0.598**±0.017 | **0.459**±0.007 | **0.529**±0.005 |
| *All-atom representation* | | | | | | | |
| SchNet | 0.427±0.003 | 0.603±0.007 | 0.472±0.007 | 0.501±0.008 | 0.850±0.015 | 0.849±0.003 | 0.849±0.009 |
| TFN | 0.400±0.001 | 0.597±0.007 | 0.478±0.006 | 0.491±0.007 | 0.817±0.014 | 0.820±0.005 | 0.819±0.006 |
| EGNN | 0.426±0.003 | 0.602±0.007 | 0.471±0.006 | 0.498±0.007 | 0.801±0.023 | 0.802±0.001 | 0.802±0.017 |
| FastEGNN | 0.423±0.003 | 0.604±0.007 | 0.489±0.007 | 0.505±0.009 | 0.846±0.003 | 0.884±0.005 | 0.865±0.004 |
| LEFTNet | 0.425±0.003 | 0.607±0.007 | 0.486±0.007 | 0.506±0.009 | 0.850±0.004 | 0.886±0.002 | 0.868±0.003 |
| TMD-ET | 0.417±0.003 | 0.594±0.007 | 0.471±0.007 | 0.494±0.009 | 0.850±0.004 | 0.860±0.000 | 0.855±0.002 |
| SE(3)-Trans. | | out-of-memory | | | | out-of-memory | |
| VNT | | out-of-memory | | | | out-of-memory | |
| Equiformer | | out-of-memory | | | | out-of-memory | |
| **SE(3)-Hyena** | **0.281**±0.004 | **0.414**±0.007 | **0.324**±0.002 | **0.339**±0.004 | **0.623**±0.018 | **0.468**±0.008 | **0.546**±0.006 |

multi-task manner using average RMSE as the loss function. For detailed training parameters, model architectures, and an ablation study on SE(3)-Hyena operations, see Appendix 6.3.

**Results** Table 1 shows average and individual label RMSE for each dataset and property. For backbone representation models, we did not observe any major differences between evaluated local context models, with EGNN and LEFTNet slightly outperforming the rest of the local context models, but still significantly underperformed compared to global context methods. This indicates the crucial role of global geometric context. Also, we observed that EGNN performed on par or slightly better than its modified version FastEGNN which relies on virtual nodes. We hypothesize this may step from the center of mass initialization of virtual nodes in FastEGNN that may be suboptimal for molecules with a strong sequence prior such as RNA, and more advanced virtual node initialization strategies may alleviate this issue.

Among global context methods, SE(3)-Hyena performs the best, considerably outperforming the second-best Vector Neuron Transformer on both datasets. We hypothesize that such significant improvement is due to the mechanism of alternating global and local context used in the SE(3)-Hyena operator. Our ablation study (Appendix 6.4) further confirms the critical role of alternating global and local context in the SE(3)-Hyena operator. In addition, long convolutions are naturally well suited for molecules with a strong sequence prior as they inherently discriminate different token orderings due to the permutation symmetry breaking of a convolution.

In the all-atom representation case, the SE(3)-Hyena demonstrates its key advantage of improved computational efficiency, being the only global context method that does not run out of memory on large RNA atomic systems. Notably, for Open Vaccine datasets, the all-atom representation performs considerably better than if only the backbone is used. This further highlights the necessity of all-atom representation for accurate in-silico modeling of biological systems.

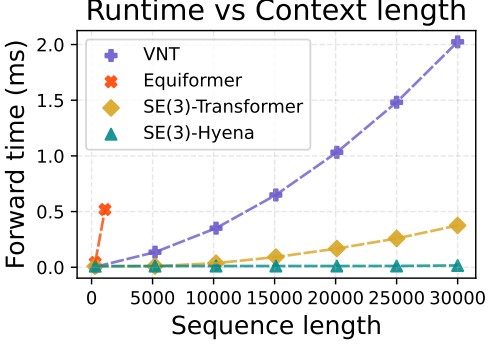 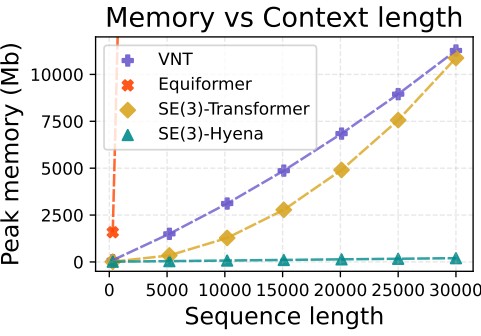

Figure 6: **Left:** Forward runtime comparison. SE(3)-Hyena scales sub-quadratically and achieves a considerable speedup compared to other globa context methods when processing long sequences. **Right:** Peak GPU memory utilization for the SE(3)-Hyena is the most efficient for long sequences.

### 4.3 Runtime and memory benchmarks

We benchmark the runtime and peak memory consumption for a forward pass of SE(3)-Hyena against other equivariant global context models: Vector Neuron Transformer (Assaad et al., 2022), Equiformer (Liao & Smidt, 2023) and SE(3)-Transformer (Appendix 6.7). Similar to Dao et al. (2022); Poli et al. (2023), we use sample sequences, and we test a one-layer model for the runtime and peak memory consumption. We record all runtimes on NVIDIA A10G GPU with CUDA 12.2.

The comparison is reported in Figure 6, showing forward pass time in milliseconds and peak GPU memory utilization in megabytes. SE(3)-Hyena easily scales to longer sequences whereas VNT, Equiformer, and SE(3)-Transformer are from $20\times$ to $120\times$ slower for a sequence length of $30k$ tokens. Regarding memory usage, our SE(3)-Hyena requires $50\times$ less GPU memory than self-attention-based methods for sequence length of $30k$ tokens. Moreover, we observed that when the Vector Neuron Transformer runs out of memory on $> 37k$ tokens, *our model supports up to $2.7M$ tokens on a single GPU allowing for $72\times$ longer geometric context*. This memory efficiency is attributed to the FFT long convolution that avoids materializing a quadratic self-attention matrix.

## 5 Conclusions

We introduced the SE(3)-Hyena operator being, to the best of our knowledge, the first equivariant long-convolutional model with sub-quadratic complexity for global geometric context. Through experiments on the dynamical system, novel equivariant associative recall task and all-atom RNA property prediction, we demonstrated that equivariant long-convolutional models can perform competitively to the equivariant self-attention while requiring a fraction of the computational and memory cost of transformers for long context modeling. Our scalable equivariant model efficiently captures the global context, highlighting its potential for a multitude of future applications in various domains.

**Limitations and Future work** This work introduces a novel approach for equivariant modeling of global geometric context at scale, with initial experiments designed to validate the fundamental principles of our method. While these experiments confirm the key advantages of our approach, they represent the first step of a comprehensive experimental analysis that is necessary to uncover the model's capabilities across a wider range of real-world tasks. On a technical side, an interesting direction for future improvement is to adapt the vector convolution to function across arbitrary dimensions as it currently relies on a cross product, which is only feasible in 3 and 7 dimensions (Massey, 1983). Also, our method relies on the FFT convolution that is not permutation equivariant. While this is not a significant limitation in domains such as molecular systems, where canonical ordering is typically available (Jochum & Gasteiger, 1977), it becomes critical in fields like point cloud processing, where establishing a canonical order is challenging. Enhancing the long-convolutional framework to incorporate permutation equivariance or learning this geometric constraint from the data could offer substantial advantages in these areas. Also, as the first step, in this work, we have only considered EGNN-like equivariant network as a projection function. Using more advanced equivariant models as the projection layer could further improve the performance.

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

# 6 APPENDIX

## 6.1 RELATED WORK

**Equivariance**  Equivariance to group transformations, particularly rotations and translations in 3D, is crucial for modeling physical systems Zhang et al. (2023). Schütt et al. (2017) condition continuous convolutional filters on relative distances to build model invariant to rotations and translations. Thomas et al. (2018); Fuchs et al. (2020); Brandstetter et al. (2021); Liao & Smidt (2023); Bekkers et al. (2023) utilize spherical harmonics as a steerable basis which enables equivariance between higher-order representations. Since computing spherical harmonics can be expensive, Jing et al. (2021b;a); Satorras et al. (2021); Deng et al. (2021) focus on directly updating vector-valued features to maintain equivariance, while Zhdanov et al. (2024a) employ another equivariant network to implicitly parameterize steerable kernels. Another recent line of work Ruhe et al. (2023a;b); Brehmer et al. (2023); Zhdanov et al. (2024b) employs geometric algebra representation which natively provides a flexible framework for processing symmetries in the data Dorst et al. (2009).

While these works focus on how to build equivariance into a neural network, in this paper we focus on efficient equivariance to model global geometric contexts at scale.

**Modeling geometric context**  Various strategies are employed to process context information in geometric data. Convolutional methods aggregate context linearly within a local neighborhood, guided by either a graph topology Kipf & Welling (2016) or spatial relations in geometric graphs Schütt et al. (2017); Wu et al. (2019); Thomas et al. (2018). Message-passing framework Gilmer et al. (2017) generalizes convolutions, facilitating the exchange of nonlinear messages between nodes with learnable message functions. These approaches are favored for their simplicity, balanced computational demands, and expressiveness Wu et al. (2020). However, they are limited to local interactions and are known to suffer from oversmoothing Rusch et al. (2023). This hinders building deep message-passing networks capable of encompassing a global geometric context in a receptive field. To address these limitations, a recent line of work focuses on architectural modifications in form of virtual nodes (Zhang et al., 2024) or using node-to-mesh message passing (Wang et al., 2024b) to enhance the capability of a model to handle long-range interactions. Being effective in a certain scenarios, these methods are constrained by their use of low-rank approximations for global interactions, limiting their expressivity when the interactions do to adhere to low-rank behavior. Another line of work adopts self-attention mechanisms for graph Yang et al. (2021); Kreuzer et al. (2021); Kim et al. (2022); Rampášek et al. (2022) and geometric graph Fuchs et al. (2020); Liao & Smidt (2023); Brehmer et al. (2023) data, outperforming convolutional and message-passing approaches. Yet, the quadratic computational cost of self-attention poses significant challenges when modeling large-scale physical systems. Along the same lines,

In this work, we aim to develop a method for global geometric context processing with sub-quadratic computational complexity.

**State-space and long-convolutional models**  The quadratic computational complexity of self-attention has driven the exploration of alternatives for modeling long context. Structured state-space models Gu et al. (2021b) have emerged as a promising alternative, integrating recurrent and convolutional mechanisms within a single framework. These models enable parallelized training in a convolutional mode and maintain linear complexity with respect to sequence length in a recurrent mode. Models like S4 Gu et al. (2021a), H3 Fu et al. (2022), and Mamba Gu & Dao (2023); Li et al. (2024) have consistently matched or exceeded transformer performance in diverse tasks such as genomics Schiff et al. (2024), long-range language Wang et al. (2024a), and vision tasks Zhu et al. (2024). Concurrently, another line of work integrates long-convolutional framework with implicit filters Sitzmann et al. (2020); Romero et al. (2021); Zhdanov et al. (2024a) to capture global sequence context. The implicit filter formulation allows for data-controlled filtering similar to transformers, while FFT-based long convolution enables global context aggregation in sub-quadratic time Poli et al. (2023). Such models have shown competitive performance comparable to state-space and transformer architectures in time-series modeling Romero et al. (2021), genomics Nguyen et al. (2024), and vision tasks Poli et al. (2023).

Although state-space and long-convolutional methods dramatically reduced the computational costs associated with processing long sequences, their application to geometric data requiring equivariance

remains unexplored. In this work, we adapt the recently proposed Hyena operator Poli et al. (2023) to incorporate SE(3) equivariance. To the best of our knowledge, this is the first equivariant long-convolutional model that can process global geometric contexts with sub-quadratic memory and time requirements.

## 6.2 DYNAMICAL SYSTEM EXPERIMENT

We evaluate SE(3)-Hyena on forecasting trajectories in a dynamical system of 5 charged particles, a task requiring equivariance to rotations and translations. The objective is to predict future particle positions given initial positions and velocities.

We use the standard n-body benchmark from Fuchs et al. (2020); Satorras et al. (2021). Input features are initial positions $\mathbf{p}_0 \in \mathbb{R}^{5 \times 3}$ and velocities $\mathbf{v}_0 \in \mathbb{R}^{5 \times 3}$, with positions after 1000 timesteps $\mathbf{p}_{1000} \in \mathbb{R}^{5 \times 3}$ as labels. We compare SE(3)-Hyena against established equivariant message passing methods: Tensor Field Network (Thomas et al., 2018), Radial Field Network(Köhler et al., 2019), EGNN (Satorras et al., 2021), and SE(3)-Transformer (MPNN) (Fuchs et al., 2020), SAKE (Wang & Chodera, 2023), SEGNO (Liu et al., 2024), SEGNN (Brandstetter et al., 2021), CEGNN (Ruhe et al., 2023a). Additionally, we include the SE(3)-Transformer variant where we replace SE(3)-Hyena's long convolutions with equivariant vector self-attention for global context aggregation.

Table 2 shows that SE(3)-Hyena achieves the performance on par with the equivariant self-attention SE(3)-Transformer (Appendix 6.7) baseline, indicating that the designed equivariant operator is on par with the self-attention module given

Table 2: MSE of estimated trajectories in the N-body system experiment.

| Method | MSE |
|---|---|
| Linear | 0.0819 |
| SE(3)-Transformer (MPNN) | 0.0244 |
| Tensor Field Network | 0.0155 |
| Radial Field Network | 0.0104 |
| EGNN | 0.0071 |
| SAKE | 0.0049 |
| SEGNO | 0.0043 |
| SEGNN | 0.0043 |
| CEGNN | **0.0039** |
| SE(3)-Transformer | 0.0058 |
| SE(3)-Hyena | 0.0059 |

the rest of the architecture is unchanged. At the same time, recent local context baselines such as SAKE (Wang & Chodera, 2023), SEGNO (Liu et al., 2024), SEGNN (Brandstetter et al., 2021) and CEGNN (Ruhe et al., 2023a) demonstrate the superior performance. These local context methods are optimized for high accuracy on small geometric systems, rather than for computational efficiency. This renders them efficient for small-scale N-body task but inapplicable for large-scale geometric systems, due to compute and memory requirements.

## 6.3 EXPERIMENT DETAILS

### 6.3.1 N-BODY DYNAMICAL SYSTEM

**Model and training** The initial positions and velocities serve as the input to the equivariant branch, and one-hot encoded charges are used as the input to the invariant branch. The SE(3)-Hyena model consists of 2 SE(3)-Hyena operators followed by one output equivariant MLP. The equivariant transformer baseline consists of 2 vector self-attention blocks and one output equivariant MLP. The hidden dimension for both models is set to 32. We also use the linear baseline model which is simply a linear motion equitation $\mathbf{p}_t = \mathbf{p}_0 + t \cdot \mathbf{v}_0$. The models are trained for 10000 epochs with a batch size of 100. We use Adam optimizer with a learning rate set to 0.0001 and weight decay of 0.00001. The mean squared error is used as the loss function.

### 6.3.2 EQUIVARIANT ASSOCIATIVE RECALL

**Datasets** We set the training data size to 2600, and validation and test set size to 200 sequences each. The vocabulary size is set to 4 for both random and fixed versions. The vocabularies are generated as follows: one key-value bigram consists of two vectors with orientations sampled as random unit 3D vectors from an isotropic normal distribution, and with magnitudes randomly sampled from a uniform distribution in a range of $[1, \texttt{vocab\_size}]$. We generate datasets with various sequence lengths from $2^5$ to $2^8$ tokens. When generating sequences from a vocabulary, the last token is a key that corresponds to a target value. We additionally constraint the generation so the target key-value pair is present in a sequence at least once.

**Models** The vector tokens are used as the input to the equivariant branch, while for the invariant branch, we use positional encoding features Vaswani et al. (2017) of dimension 16 as the input. The SE(3)-Hyena model consists of 2 SE(3)-Hyena operators followed by a global pooling of equivariant features and one equivariant output projection. In SE(3)-Hyena operator the hidden dimension is set to $64$. This ends up in approximately $400k$ trainable parameters. Similarly, the equivariant transformer consists of 3 SE(3)-equivariant vector self-attention blocks followed by a global pooling and an equivariant output projection. The hidden dimension is kept similar to SE(3)-Hyena which results in a nearly identical number of trainable parameters. The local neighborhood of each token in EGNN projection layer is defined as its one immediate preceding and one subsequent tokens. Non-equivariant baselines consist of 3 standard Hyena or self-attention operations but with hidden dimensions chosen to match the number of parameters of equivariant models.

**Training** We train all models for $400$ epochs with a batch size of $8$. We employ Adam optimizer Kingma & Ba (2014) with an initial learning rate of $0.001$ and cosine learning rate scheduler Loshchilov & Hutter (2016) with 10 epochs of linear warmup. The weight decay is set to $0.00001$. Mean squared error is used as a loss function. For the fixed vocabulary experiment, we apply random rotation augmentation on sequences in the training batch, for the random vocabulary variant this is not necessary as sequences already appear in arbitrary orientations. Final models are selected based on the best validation loss.

### 6.3.3 ALL-ATOM RNA

**Datasets** For the OpenVaccine COVID-19 dataset that is availble with signal-to-noise ratio (SNR) annotations, we filter out sequences with a SNR lower than 1. The resulting dataset contains mRNA sequences up to 130 nucleitodes which equates to 7800 atoms. The complete Ribonanza dataset contains approximately 168k sequences across 17 sub-datasets. From these, we select the sub-dataset with the label distribution closest to a normal distribution, referred to as Ribonanza 2k. Both Open Vaccine and Ribinanza-2k contain properties that, among others, depend on the connection between RNA nucleotides. Thus, solving these two datasets additionally requires learning the hierarchy of nucleotide associations from raw all-atom structures, a task often requiring specialized methods Anonymous (2024b). We utilize state-of-the-art 3D structure prediction tool RhoFold Shen et al. (2022) to get 3D structures for all datasets. Compared to other existing 3D structure prediction tools which can take hours to run for even single sequences, RhoFold is reported to be more accurate and fast to run (usually few seconds to a minute per sequence) and has been used in prior works as the tool of choice (He et al., 2024).

For the 3D datasets, each RNA molecule is represented as a graph where nodes correspond to individual atoms, and edges are defined by the point cloud representation, capturing the spatial arrangement of these atoms. The node features have a dimension of 6, encoding the atomic number as an integer and including a one-hot encoding for the elements ('Hydrogen', 'Carbon', 'Nitrogen', 'Oxygen', 'Phosphorus').

**Models and training** We use atom identities as invariant features and 3D coordinates as equivariant features. We supplement invariant features with nucleotide identities to indicate each atom's parent nucleotide. The network processes an all-atom atomic cloud representation of RNA and outputs stability and degradation profiles for each nucleotide. To map from atom to nucleotide level, we sum atom features within each nucleotide in the last layer before making a prediction. We utilize the SE(3)-Hyena model that consists of 2 SE(3)-Hyena operators together with input and output MLP projections, which in total equates to 5 layers. Consequently, we compare it against 5-layer baseline SchNet, TFN, EGNN, SE(3)-Transformer, VNT. For Equiformer we only use 3 equivariant self-attention layers due to its high computational cost. We use the hidden dimension of $84$ for the SE(3)-Hyena model and we choose the hidden dimension of other models to equate the number of parameters. To further speed up the training of the SE(3)-Hyena model, we only use 32 (set as hyperparameters) closest neighbors to encode local context, instead of using all neighbors within the radius threshold. The radius threshold for local methods is set to $1.6\mathring{A}$ (the maximum radius permitted by memory requirement) for all-atom representation and to $4\mathring{A}$ for backbone representations.

Following Wayment-Steele et al. (2022), we train all models in a multi-task manner, predicting all properties simultaneously using average root mean squared error (RMSE) as the loss function. Training runs for approximately 200 epochs with a batch size of 16. We employ the Adam optimizer

(Kingma & Ba, 2014) with an initial learning rate of 0.001, using a cosine learning rate schedule with a 10-epoch of linear warm-up. Weight decay is set to 0.0005.

## 6.4 ABLATION STUDY

We provide the ablation study on the modules used in SE(3)-Hyena operator below in Table 3. We test the impact of various modules for the model used in RNA property prediction Experiment 4.2. We follow the same experimental setup with the same models as in the main experiment but we train the models on one common random seed to reduce training and evaluation time.

Table 3: Ablation study results. Checkmarks indicate the presence of a component. Lower RMSE values indicate better performance. The best result is highlighted in bold.

| Operation | | | | | Avg. RMSE | |
|---|---|---|---|---|---|---|
| Local context | Global context | Gating | KV norm | Geometric convolution | Open Vaccine Covid-19 | Ribonanza-2k |
| ✓ | ✓ | QK | ✓ | ✓ | **0.339** | **0.544** |
| ✓ | ✓ | K | ✓ | ✓ | 0.345 | 0.547 |
| ✓ | ✓ | | ✓ | ✓ | 0.449 | 0.551 |
| | ✓ | QK | ✓ | ✓ | 0.450 | 0.614 |
| ✓ | | QK | ✓ | ✓ | 0.421 | 0.545 |
| ✓ | ✓ | QK | | ✓ | NaN | NaN |
| ✓ | ✓ | QK | ✓ | | 0.349 | 0.557 |

As can be seen from Table 3, the optimal configuration for the SE(3)-Hyena operator includes using both local and global context with the gating applied on top of the features resulting from geometric long convolution (QK). Note that using geometric long convolution with scalar-vector interaction improves the performance of the model compared to using scalar and vector long convolutions separately. Similarly, using local context in SE(3)-Hyena yields a significant improvement in the model's performance. Also, we observe that using key-value normalization (KV-norm) is critical to ensure stable convergence of the model.

Overall, the ablation study reveals that *alternating global and local context, key-value normalization, and geometric long convolution* are essential for SE(3)-Hyena stability and performance.

## 6.5 DETAILS ON THE SE(3)-HYENA COMPONENTS

### 6.5.1 GEOMETRIC LONG CONVOLUTION BY SCALAR AND VECTOR LONG CONVOLUTIONS

We provide a detailed explanation of how the geometric long convolution is implemented using scalar and vector long convolutions. Recall that the geometric long convolution combines scalar and vector convolutions to enable scalar-vector interactions. Given input scalar-vector tuples $(\alpha_1, \mathbf{r}_1)$ and $(\alpha_2, \mathbf{r}_2) \in \mathbb{R} \times \mathbb{R}^3$, we defined the mapping to output scalar-vector tuples $(\alpha_3, \mathbf{r}_3)$ as:

$$\alpha_3 = \lambda_1 \alpha_1 \alpha_2 + \lambda_2 \mathbf{r}_1^T \mathbf{r}_2$$
$$\mathbf{r}_3 = \lambda_3 \alpha_1 \mathbf{r}_2 + \lambda_4 \alpha_2 \mathbf{r}_1 + \lambda_5 (\mathbf{r}_1 \times \mathbf{r}_2)$$

To implement this in a convolutional manner, we decompose the geometric long convolution into a series of scalar and vector long convolutions. For clarity, we omit the token index $i$ in the following equations, treating $\alpha$ and $\mathbf{r}$ as scalar and vector-valued signals of $N$ tokens, respectively.

**Scalar output computation** The scalar output $\alpha_3$ involves two terms: (i) scalar-scalar product: $\lambda_1 \alpha_1 \alpha_2$, and (ii) vector dot product: $\lambda_2 \mathbf{r}_1^T \mathbf{r}2$. The scalar-scalar term can be directly computed using the scalar long convolution as defined in Equation 2: $\alpha_1 \circledast \alpha_2 = \mathbf{F}^H \text{diag}(\mathbf{F}\alpha_2)\mathbf{F}\alpha_1$. The vector dot product can be decomposed into three scalar convolutions, one for each dimension:

$(\mathbf{r}_1^T \mathbf{r}_2)_i = \sum_{d=1}^{3} (r_1[d] \circledast r_2[d])i$ where $r_1[d]$ and $r_2[d]$ represent the $d$-th component of vectors $\mathbf{r}_1$ and $\mathbf{r}_2$, respectively. With this, the scalar part output of the geometric long convolution can be written as:

$$\alpha_3 = \lambda_1(\alpha_1 \circledast \alpha_2) + \lambda_2 \sum_{d=1}^{3} (r_1[d] \circledast r_2[d]) \tag{5}$$

**Vector output computation** The vector output $\mathbf{r}_3$ involves three terms: (i) scalar-vector product: $\lambda_3 \alpha_1 \mathbf{r}_2$, (ii) scalar-vector product: $\lambda_4 \alpha_2 \mathbf{r}_1$, and (iii) vector cross product: $\lambda_5(\mathbf{r}_1 \times \mathbf{r}_2)$. Both scalar-vector product terms can be computed using scalar convolutions for each vector component as $\alpha_1 \circledast r_2[d]$ and $\alpha_2 \circledast r_1[d]$ for $d$-th component of the resulting vector. The vector cross product part can be computed using the vector long convolution as defined in Equation 3 of the main text. With this, the vector part output of the geometric long convolution can be written as:

$$\mathbf{r}_3 = \lambda_3 \bigsqcup_{d=1}^{3} (\alpha_1 \circledast \mathbf{r}_2[d]) + \lambda_4 \bigsqcup_{d=1}^{3} (\alpha_2 \circledast \mathbf{r}_1[d]) + \lambda_5(\mathbf{r}_1 \circledast_{\times} \mathbf{r}_2) \tag{6}$$

where $\bigsqcup_{d=1}^{3}$ denotes concatenation along the $d$-th component.

### 6.5.2 PROOF OF EQUIVARIANCE

**Projection function** Our framework allows any equivariant network as a projection function, making equivariance a prerequisite. We use a modified EGNN (Satorras et al., 2021) with global messages $\mathbf{m}_{ij}^{glob} = \varphi_g(\mathbf{f}_i, \mathbf{h}_j, \log(1 + \|\mathbf{x}_i - \mathbf{g}_j\|_2))$. This maintains E(n)-equivariance if global context tokens $\mathbf{g}_j$ are equivariant. The proof follows EGNN's, as $\mathbf{f}_i$ and $\mathbf{h}_j$ are invariant, and $\mathbf{f}_i, \mathbf{h}_j$ are invariant tokens and $\|(\mathbf{R}\mathbf{x}_i + t) - (\mathbf{R}\mathbf{g}_j + t)\|_2 = \|\mathbf{R}(\mathbf{x}_i - \mathbf{g}_j)\|_2 = \|\mathbf{x}_i - \mathbf{g}_j\|_2$.

**Global context tokens** Global context tokens provide information about the overall structure of geometric graphs. While representing a compressed snapshot, they make scalar and vector feature updates aware of their global position. We create this snapshot by taking a weighted average of scalar or vector tokens across all input tokens. Given input tokens $\{\mathbf{x}_i, \mathbf{f}_i\}_{i=1}^{N}$, a set of $G$ geometric tokens $\{\mathbf{g}_j, \mathbf{h}_j\}_{j=1}^{G}$ is obtained as $\mathbf{g}_j = C_j^{-1} \sum_{i=1}^{N} \omega_{ij} \mathbf{x}_i$ for vector tokens and $\mathbf{h}_j = C_j^{-1} \sum_{i=1}^{N} \omega_{ij} \mathbf{f}_i$ for scalar tokens where $C_j = \sum_{i=1}^{N} \omega_{ij}$. The weighting factor $\omega_{ij}$ is a learned scalar representing the contribution of the $i$-th input token to the $j$-th global context token. To detach the learning process from specific sequence length requirements, we employ a small SIREN network (Sitzmann et al., 2020) to predict $\omega_{ij}$ for each position.

**Global context tokens** We next proof the equivariance of global context token mechanism. Recall that given input tokens $\{\mathbf{x}_i, \mathbf{f}_i\}_{i=1}^{N}$, a set of $G$ geometric tokens $\{\mathbf{g}_j, \mathbf{h}_j\}_{j=1}^{G}$ is obtained as $\mathbf{g}_j = C_j^{-1} \sum_{i=1}^{N} \omega_{ij} \mathbf{x}_i$ for vector tokens and $\mathbf{h}_j = C_j^{-1} \sum_{i=1}^{N} \omega_{ij} \mathbf{f}_i$ for scalar tokens where $C_j = \sum_{i=1}^{N} \omega_{ij}$ with $\omega_{ij}$ being learnable weights. Firstly, $\mathbf{h}_j$ maintain invariance as a function quantities $\mathbf{f}_i$ and $\omega_{ij}$ which by design only depend on the ordinal position of the token in a sequence. For the equivariance part:

$$C_j^{-1} \sum_{i=1}^{N} \omega_{ij}(\mathbf{R}\mathbf{x}_i + t) = C_j^{-1} \sum_{i=1}^{N} \omega_{ij} \mathbf{R}\mathbf{x}_i + C_j^{-1} \sum_{i=1}^{N} \omega_{ij} t$$

$$= \mathbf{R}\left(C_j^{-1} \sum_{i=1}^{N} \omega_{ij} \mathbf{x}_i\right) + \left(\sum_{i=1}^{N} \omega_{ij}\right)^{-1}\left(\sum_{i=1}^{N} \omega_{ij}\right) t$$

$$= \mathbf{R}\mathbf{g}_j + t$$

**Vector long convolution**  Given a vector signal consisting of $N$ vector tokens $\mathbf{q}^{eqv} \in \mathbb{R}^{N \times 3}$ and a vector kernel $\mathbf{k}^{eqv} \in \mathbb{R}^{N \times 3}$, recall that $i$-th element of the result if the vector long-convolution is defined as $\sum_{j=1}^{N} \mathbf{q}_i^{eqv} \times \mathbf{k}_{j-i}^{eqv}$. Since a cross product is already equivariant to rotations, the whole vector convolution is also equivariant to rotations provided the queries $\mathbf{q}^{eqv}$ and the keys $\mathbf{k}^{eqv}$ are rotated accordingly (which is guaranteed when the projection function is equivariant). Formally:

$$\sum_{j=1}^{N} \mathbf{R}\mathbf{q}_i^{eqv} \times \mathbf{R}\mathbf{k}_{j-i}^{eqv} = \sum_{j=1}^{N} \mathbf{R}(\mathbf{q}_i^{eqv} \times \mathbf{k}_{j-i}^{eqv}) = \mathbf{R}\sum_{j=1}^{N} \mathbf{q}_i^{eqv} \times \mathbf{k}_{j-i}^{eqv}$$

This provides equivariance to the SO(3). For the SE(3) equivariance, we can first center the input tokens relative to their center of mass, apply the convolution, then uncenter the result.

**Geometric long convolution**  The geometric long convolution combines scalar and vector convolutions for scalar-vector interactions. We prove its equivariance by examining each term under transformations. For the scalar output: (i) $\alpha_1 \circledast \alpha_2$ is invariant as it involves only scalar quantities; (ii) $\sum_{d=1}^{3}(r_1[d] \circledast r_2[d])$ is rotation-invariant as it reassembles a dot product which is invariant to rotations. For the vector output: (i) Terms $\bigsqcup_{d=1}^{3}(\alpha_1 \circledast r_2[d])$ and $\bigsqcup_{d=1}^{3}(\alpha_2 \circledast r_1[d])$ transform equivariantly under rotation $\mathbf{R}$ because $\mathbf{R}(\alpha \circledast \mathbf{r}) = \alpha \circledast (\mathbf{R}\mathbf{r})$ for any scalar $\alpha$ and vector $\mathbf{r}$, as convolution distributes over vector components; (ii) $\mathbf{r}_1 \circledast_{\times} \mathbf{r}_2$ is equivariant as previously proven for vector long convolution. This establishes SO(3) equivariance. For SE(3) equivariance, we center input tokens to their center of mass before convolution and uncenter afterwards, making the operation translation-equivariant.

**Selective gating**  The gating function $g : \mathbb{R}^3 \times \mathbb{R}^d \to [0, 1]$ is composed of an equivariant projection layer, a linear layer, and a sigmoid activation. Since the projection layer is equivariant (as proven earlier), and both the linear layer and sigmoid operate only on top of invariant features, the gating function itself is invariant to SE(3) transformations. For scalar tokens, the gating operation $m_i \mathbf{u}_i^{inv}$ preserves invariance as it's a product of invariant terms. For vector tokens, under rotation $\mathbf{R}$, we have $m_i(\mathbf{R}\mathbf{u}_i^{eqv}) = \mathbf{R}(m_i \mathbf{u}_i^{eqv})$ since $m_i$ is a scalar. For translations, centering and uncentering steps ensure translation equivariance, obtaining the SE(3)-equivariance.

**Key-Value normalization**  We prove that the key-value normalization preserves SO(3) equivariance but not SE(3) equivariance. We normalize keys and values to unit norm: $\|\mathbf{k}\|_2 = \|\mathbf{v}\|_2 = 1$. For scalar keys and values, this normalization is invariant to rotations as it operates on top of invariant quantities. For vector keys and values, under rotation $\mathbf{R}$, we have $\|\mathbf{R}\mathbf{k}\|_2 = \|\mathbf{k}\|_2 = 1$ and $\|\mathbf{R}\mathbf{v}\|_2 = \|\mathbf{v}\|_2 = 1$. The normalized vectors transform as $\mathbf{k}/\|\mathbf{k}\|_2 \to \mathbf{R}\mathbf{k}/\|\mathbf{R}\mathbf{k}\|_2 = \mathbf{R}(\mathbf{k}/\|\mathbf{k}\|_2)$, and similarly for $\mathbf{v}$. Thus, the normalization commutes with rotations, providing the SO(3) equivariance. For translations, centering and uncentering steps ensure translation equivariance, obtaining the SE(3)-equivariance.

## 6.6  EXTENSION TO HIGHER-ORDER REPRESENTATION

To further encourage the research towards equivariant long-convolutional models for geometric systems, we outline a blueprint of the potential extension of SE(3)-Hyena-like architecture to accommodate higher-order representations. We consider two possibilities: (i) employing spherical harmonics-based steerable representation with the tensor product, and (ii) interaction with higher-order features by scalarization trick Satorras et al. (2021); Cen et al. (2024); Anonymous (2024a).

Extending the SE(3)-Hyena to higher-order steerable features requires working out steerable vector long convolution. Conceptually, the same mathematical approach can be applied but instead of vectors in and cross product, we will have steerable vectors and tensor product. Then, the long convolution can be built with tensor product and reduced to a series of FFT scalar convolutions by factoring out CG coefficients. This is the overall blueprint of one of the possible approaches. In this approach, various types of steerable vectors will interact via standard or parametrized tensor product.

**Steerable long convolution with higher-order representations**  Following the steps used in this work to derive the cross product-based vector long convolution (Section **??**), one can employ steerable

representations and tensor products to derive a long convolution for higher-order steerable vectors. Instead of vectors in $\mathbb{R}^3$ and the cross product, we use steerable vectors and the tensor product. The long convolution can then be constructed using the tensor product and reduced to a series of scalar convolutions by factoring out the Clebsch-Gordan (CG) coefficients.

Let $\mathbf{q}^{(l_1)}$ and $\mathbf{k}^{(l_2)}$ be sequences of steerable features of degrees $l_1$ and $l_2$ respectively, respectively, each with length $N$. These features transform according to the spherical harmonics representations of SO(3). The steerable long convolution can be defined as:

$$\left( \mathbf{q}^{l_1} \circledast_{\text{CG}} \mathbf{k}^{l_2} \right)^{(l)}_{i,m} = \sum_{j=1}^{N} \sum_{m_1=-l_1}^{l_1} \sum_{m_2=-l_2}^{l_2} C^{(lm)}_{(l_1 m_1)(l_2 m_2)} \mathbf{q}^{(l_1)}_{i,m_1}, \mathbf{k}^{(l_2)}_{j-i,m_2} \tag{7}$$

where $C^{(lm)}_{(l_1 m_1)(l_2 m_2)}$ denotes Clebsch-Gordan coefficients from type $l_1, l_2$ to $l$.

To efficiently compute this convolution, we can decompose it into scalar convolutions factoring out the Clebsch-Gordan coefficients, reducing the computational complexity from $O(N^2)$ to $O(N log N)$:

$$\mathbf{u}^{(l)}_{i,m} = \sum_{m_1=-l_1}^{l_1} \sum_{m_2=-l_2}^{l_2} C^{(lm)}_{(l_1 m_1)(l_2 m_2)} \left( \mathbf{q}^{(l_1)}_{m_1} \circledast \mathbf{k}^{(l_2)}_{m_2} \right)_i \tag{8}$$

*This is a direct generalization of our method*, since the cross product used in our vector long convolution corresponds to the case where $l = l_1 = l_2 = 1$, and the Clebsch-Gordan coefficients reduce to the Levi-Civita symbol.

The advantage of the fully steerable approach is that it allows interactions between features of arbitrary degrees, capturing complex geometric relationships and maintaining strict equivariance under rotations. However, this approach might be computationally expensive due to the large number of scalar convolutions and Clebsch-Gordan coefficients involved, especially for higher degrees. Potentially, this computational overhead hinders the usage of fully steerable methods for large-scale geometric systems, where efficiency is crucial. However, in this paper, we purposefully chose to focus on Cartesian vectors with the cross product due to its computational efficiency and simplicity. By restricting to type-1 representations, we reduce the computational complexity and make the method more practical for large-scale applications, while still capturing essential geometric properties through equivariant operations.

**Faster higher-order representations by scalarization** Alternatively, to efficiently model interactions between high and low order features without the heavy computational cost of fully steerable methods, we can use the scalarization trick. This approach extends the geometric long convolution framework by introducing higher-order components into the feature representations while simplifying computations by selectively considering interactions.

In this method, each feature tuple is expanded to include a higher-order part. Specifically, using notation from Section 2.2, we extend feature representation from $(\alpha, \mathbf{r})$ to $(\alpha, \mathbf{r}, \mathbf{v})$ containing scale, vector, and higher-order representation respectively. When performing the convolution between two such feature tuples, we drop the higher-order to higher-order interaction terms. With this, we can focus on interactions that involve higher-order to low-order order and low-order to low-order interactions only, effectively scalarizing the processing of higher-order features.

By excluding higher-order to higher-order interactions, we significantly reduce computational complexity. The higher-order features still contribute to the model through their interactions with scalar parts, allowing the network to capture important geometric information without the full computational overhead associated with processing all higher-order interactions.

### 6.6.1 PYTORCH IMPLEMENTATION

Equivariant vector long convolution is an essential building block in the SE(3)-Hyena operator. It can be used as is to aggregate global geometric context across vector tokens or can be combined with standard scalar long convolution into geometric long convolution. Here, we provide Pytorch

implementation for the rotation-equivariant (without centering) vector long convolution in Code 1. Further SE(3)-equivariance is obtained by centering the inputs and uncentering the output of the long convolution.

```python
class VectorLongConv(nn.Module):
    def __init__(self):
        super(VectorLongConv, self).__init__()

        # L cross-prod tensor factorized:
        expand_vec_mat = torch.FloatTensor([[1,  0,  0],
                                            [1,  0,  0],
                                            [0,  1,  0],
                                            [0,  1,  0],
                                            [0,  0,  1],
                                            [0,  0,  1]]).view(1,1,1,6,3)
        self.register_buffer("expand_vec_mat", expand_vec_mat, persistent=False)

        # H cross-prod tensor factorized:
        cross_expand_vec_mat = torch.FloatTensor([[0 ,  1,  0],
                                                  [0 ,  0,-1],
                                                  [-1,  0,  0],
                                                  [0 ,  0,  1],
                                                  [1 ,  0,  0],
                                                  [0 ,-1,  0]]).view(1,1,1,6,3)
        self.register_buffer("cross_expand_vec_mat", cross_expand_vec_mat, persistent=False)

        # P cross-prod tensor factorized:
        cross_sum_mat = torch.FloatTensor([[0,0,0,1,0,1],
                                           [0,1,0,0,1,0],
                                           [1,0,1,0,0,0]]).view(1,1,1,3,6)
        self.register_buffer("cross_sum_mat", cross_sum_mat, persistent=False)

    def conv_fft(self, x, k):
        N = x.shape[-2]
        fft_x = torch.fft.rfft(x, n=N, dim=-2)
        fft_k = torch.fft.rfft(k, n=N, dim=-2)
        conv_xk = torch.fft.irfft(fft_x*fft_k, norm="backward", n=N, dim=-2)
        return conv_xk

    def forward(self, x, k):

        # batch, channel, sequence length, 3
        B,C,N,_ = x.shape

        # expand aux matrices
        expand_vec_mat = self.expand_vec_mat.expand(B,C,N,6,3)
        cross_expand_vec_mat = self.cross_expand_vec_mat.expand(B,C,N,6,3)
        cross_sum_mat = self.cross_sum_mat.expand(B,C,N,3,6)

        # expand inputs and matmul
        expanded_x = torch.matmul(expand_vec_mat, x.unsqueeze(-1)).squeeze(-1)
        cross_expanded_k = torch.matmul(cross_expand_vec_mat, k.unsqueeze(-1)).squeeze(-1)

        # fft conv
        fft_conv_xk = self.conv_fft(expanded_x, cross_expanded_k)
        reduced_fft_conv_cd = torch.matmul(cross_sum_mat, fft_conv_xk.unsqueeze(-1)).squeeze(-1)   / N

        return reduced_fft_conv_cd
```

Code 1: Pytorch implementation of the equivariant vector long convolution.

## 6.7 DETAILS ON SE(3)-TRANSFORMER BASELINE

Since SE(3)-Hyena utilizes long convolutions based on cross products, we design cross product equivariant self-attention to align the global geometric context aggregation for Hyena and transformer models for more direct comparison. With this, we use a similar architecture to SE(3)-Hyena (i.e. input projection, QKV-projection, output-projection) but instead of long-convolution, we utilize cross product equivariant self-attention.

**Cross product vector self-attention** Consider sequences of $N$ vector query, key, and value tokens denoted as $\mathbf{q}, \mathbf{k}, \mathbf{v} \in \mathbb{R}^{N \times 3}$. We construct a query-key cross product tensor $\mathbf{C} \in \mathbb{R}^{N \times N \times 3}$ where each element $\mathbf{C}_{ij} = \mathbf{q}_i \times \mathbf{k}_j$, or using Levi-Civita notation as in Eq. 4, $\mathbf{C}_{ij}[l] = \varepsilon_{lhp}\mathbf{q}_i[h]\mathbf{k}_j[p]$. To

integrate a softmax selection mechanism, as in standard self-attention, we first compute a matrix $\eta(\mathbf{C}) \in \mathbb{R}^{N \times N}$ containing the $L_2$ norms of all cross products, specifically $\eta(\mathbf{C})_{ij} = \|\mathbf{q}_i \times \mathbf{k}_j\|_2$. Applying softmax to $\eta(\mathbf{C})$ then determines the vector pairs to select from the cross product tensor. Lastly, the values $\mathbf{v}$ are cross-multiplied with the softmax-filtered cross product tensor. Overall, the equivariant vector self-attention reads as:

$$\mathbf{S} = \texttt{softmax}(\frac{1}{\sqrt{N}}\eta(\mathbf{C})) \odot \mathbf{C} \tag{9}$$

$$\mathbf{u}_i = \frac{1}{N} \sum_{j=1}^{N} \mathbf{S}_{ij} \times \mathbf{v}_j \tag{10}$$

where the softmax is applied row-wise, and $\odot$ stands for element-wise product. Consequently, $\mathbf{S} \in \mathbb{R}^{N \times N \times 3}$ represents a tensor that encapsulates a soft selection of cross products between $\mathbf{q}_i$ and $\mathbf{k}_j$. We also considered using just $\texttt{softmax}(\frac{1}{\sqrt{N}}\eta(\mathbf{C}))$ as a self-attention matrix, but early experiments indicated that the method outlined in Eq. 9 yields better results. Additionally, we found that normalizing the sum by $1/N$ in Eq. 10 further improves convergence.

Since the tensor $\mathbf{C}$ is constructed using cross products, it naturally maintains equivariance to rotations of queries and keys. Furthermore, the softmax is applied to the $L_2$ norms of the cross products making makes it rotation-invariant. Consequently, the self-attention tensor $\mathbf{S}$ is a product of rotation-invariant scalar and rotation-equivariant vector quantities, rendering it rotation-equivariant. The Eq. 10 further preserves rotation-equivariance due to the inherent equivariance of the cross product. Equivariance to translations can be achieved by initially centering the data (subtracting the center of mass) and then re-centering the resulting tokens.

