# OpenReview forum: "SE(3)-Hyena Operator for Scalable Equivariant Learning"
_ICLR.cc/2025/Conference — ICLR 2025 Conference Withdrawn Submission_

### Official Review · Reviewer_ZLS9 · 2024-10-31

**Soundness:** 1
**Presentation:** 1
**Contribution:** 2
**Rating:** 3
**Confidence:** 4

**Summary:**

The authors proposed SE(3)-Hyena operator for modeling global interaction of atomistic systems. Unlike naive SE(3)-equivariant attention, the SE(3)-equivariant operator does not require quadratic computational complexity due to the usage of FFT. The reduced complexity is well-benchmarked on toy examples.

**Strengths:**

1. Using a sub-quadratic operator for global context in 3D atomistic modeling can be potentially a good idea.

**Weaknesses:**

1. The writing should be greatly improved. See Questions below for more details.
2. Experiments in the paper are very limited. The first two experiments basically tell little about how effective a network architecture can be. The third one does not show the benefit of SE(3)-Hyena except that the proposed method takes less memory.
3. Lack of comparisons to previous works on other better-benchmarked datasets such as MD17, QM9 and so on. Overall, how effective modeling global context is remains unclear.
4. The exact proposed architecture is missing or very hard to understand. I think a better visualization covering all the details in a high-level manner can be helpful.
5. The proposed architecture only uses type-0 and type-1 vectors and can only use 1 channel for type-1 vectors. These significantly limit applying the network to slightly larger datasets.

**Questions:**

> Writing
1. Line 19: Give one or two sentences about long convolutions. Otherwise, it is hard to tell the difference from message passing.
2. Line 22: Give the name of datasets you tested so that people can better judge the scale of experiments at the beginning.
3. Line 32 -- 38: Give one or two sentences about equivariance.
4. Figure 1: I cannot tell any difference from typical self-attention, and thus the complexity O(N log N) is unclear.
5. Figure 2: The figure is too simplified without giving any detail. You should include more details about projection, geometric long convolution and gating and make them consistent with the context.
6. Line 144 -- 145: Add equation number. Also I think $\hat{x}_i$ is not defined here.
7. Line 169 -- 170: typo: "To this end,"
8. Line 191 -- 192: I don't think $F^H$ is defined.
9. Line 237 -- 239: Spherical harmonics with type-0 and type-1 vectors are the same as the representation in Cartesian space.
10. Section 3 should be reflected in figures.

---

> Experiments

1. Figure 5: Why is SE(3)-Hyena better than SE(3)-Transformer? I think some explanations would be great.
2. Line 463: Why using just 2 layers? In such small models, global context probably does not exist, resulting a potentially unfair comparison.
3. Line 473 -- 474: You can train a smaller SE(3)-Transformer to fit on GPU?
4. Please compare the proposed network with other previous works on QM9 and MD17 datasets.

---

> Reproducibility

Please submit the code when the work is under review instead of releasing upon acceptance, especially when parts of the paper are unclear and experiments are not well verified.

---

> Question

1. Is the memory complexity still $O(N^2)$? If not, please give a very short introduction to this in the paper (either in the main text or appendix).
2. Line 294 -- Line 302: The weighting factor depends on the index of input tokens. How is the permutation of tokens handled?
3. Line 269: You still use typical message passing (EGNN layer) to encode the context. Would this be the memory bottleneck instead of SE(3)-Hyena operator? If yes, I think other equivariant networks share the same memory and compute bottleneck as this work.

---

> ### Author Response · Authors · 2024-11-25
> **Author response (1/2)**
>
> We appreciate the reviewer's emphasis on the clarity of the manuscript. We next address the weaknesses and questions raised by the reviewer, grouping similar questions/comments:
>
> ### Experiments on small-scale QM9 and MD17 datasets:
> We wish to clarify that the focus of our work is on large-scale geometric systems that require modeling global context with computational efficiency. For instance, the associative recall experiment involves geometric systems with up to 2^10 tokens, while the RNA experiments require modeling systems containing 5800 to 11300 atoms. In contrast, the QM9 dataset consists of small molecules with up to 30 atoms, and the MD17 dataset contains only up to 25 atoms. These small-dimensional datasets present fundamentally different challenges and requirements to the model than what is required for global context modeling of large geometric systems. Consequently, methods optimized for QM9 or MD17 face computational limitations which renders them inapplicable for large-scale datasets. With this, we do not consider the QM9 or MD17 as representative tasks to comprehensively evaluate the capabilities of our model, and benchmark it against other methods.
>
> ### Extending experiments in the paper:
> We agree with the reviewer that the experiments in the paper can be extended to provide a more comprehensive comparison of our model against other existing equivariant models.
>
> For the RNA experiment, we have now included five additional baseline models: LEFTNet, FastEGNN, TorchMD-NET Equivariant Transformer, Equiformer, and Vector Neuron Transformer. Additionally, we revealed that using a slightly larger local context (previously the projection layer EGNN only used k=10 closest neighbors) window significantly improves the performance of our model. The updated results are presented in Table 1 of the revised paper. Our SE(3)-Hyena model performs competitively with other local and global context baselines, which we attribute to its mechanism of alternating between local and global context (quantitatively supported in Table 3, Appendix 6.4). Moreover, while global context baselines like Equiformer, VNT, and SE(3)-Transformer encounter memory limitations with all-atom representations, SE(3)-Hyena still can process large-scale all-atom representation, further improving prediction quality on the Open Vaccine dataset. We additionally include the runtime and memory comparison with new global context baselines (Figure 6) in the revised version.
>
> Also, we include other global context baselines (Vector Neuron Transformer and Equiformer) in the associative recall experiment. These baselines perform on par with SE(3)-Hyena in the fixed vocabulary setting but are significantly worse in the harder random vocabulary case where a model must learn an equivariant retrieval mechanism. We attribute the superior performance of the SE(3)-Hyena in this task to the alternating usage of both local and global context, so the model can first associate key-value bi-grams locally, and then compare those bi-grams on a whole sequence level. It demonstrates that not all equivariant models can solve the associative recall equally well. The updated results are in the revised manuscript.
>
> ### The proposed architecture only uses type-0 and type-1 vectors and can only use 1 channel for type-1 vectors. These significantly limit applying the network to slightly larger datasets.
> We kindly disagree with the reviewer that using type-0 and 1 channel type-1 limits the application of the network to larger datasets. In fact, we choose to rely only on type-0 and 1-channel type-1 representation due to computational requirements induced by multi-channel higher-order representation, and the corresponding complexity of a tensor product between them. In Figure 6 of the revised paper, we demonstrate that the higher-order method such as Equiformer rapidly runs out of memory even on moderate sequence lengths, while our methods scales efficiently to large geometric systems.
>
> ### Writing and Clarifications:
> - Figure 1. We clarified the caption of Figure 1. The O(N log N) is due to the computational complexity of the long convolution.
> - Figure 2: The goal of the figure is to provide the overall overview of the architecture with its key components: query-key-value projection, geometric long convolution, and gating, all detailed in Section 2. We would appreciate the reviewer's suggestion on which general architecture details require further elaboration and clarification in Figure 2.
> - $\hat{x}_{i}$ is the vector-part output of the SE(3)-Hyena operator defined in Equation 1, and $F^{H}$ is the standard notation for the Hermitian transpose of the DFT matrix.
> - We thank the reviewer for suggesting text clarifications. We have now absorbed them into the text where appropriate and fixed typos.

---

> ### Author Response · Authors · 2024-11-25
> **Author response (2/2)**
>
> ### Other questions:
> - Is the memory complexity still O(N^2)? No, the computational complexity is O(N log N) due to the FFT long convolution.
> - You still use typical message passing (EGNN layer) to encode the context. Would this be the memory bottleneck instead of the SE(3)-Hyena operator? Indeed, we employ a slight modification of EGNN to encode local context in the SE(3)-Hyena operator. For computational efficiency, our implementation of EGNN only operates in a small limited radius-neighborhood and only uses k (set as hyperparameter) closest neighbors to encode local context. This ensures its computational efficiency (and linear computational complexity) even for large geometric systems.
> - While representation in Cartesian space and type-0 and type-1 vectors are indeed similar in how they transform under rotation, they are not the same in a strict sense. Note that, type-0 and type-1 vectors result from additionally embedding a Cartesian vector into a Spherical Harmonics basis.
> - Why using just 2 layers in RNA experiment? This is a typo, thanks for noticing! In fact, in our architecture, we use 2 SE(3)-Hyena building blocks (Figure 2), which equates to using 6 induvidual parameteric layers. Consequently, we use 6 layers for other models and tune their hidden dimension to equate the number of parameters. We clarify this in the implementation details in the Appendix.
> - Figure 5: Why is SE(3)-Hyena better than SE(3)-Transformer? Great question! This result is consistent with the recent literature [1,2] that demonstrates that long-convolutional and state-space models can outperform self-attention on long sequences. Also, the proposed geometric long convolution allows interaction between invariant and equivariant subspaces, which is not the case in the baseline SE(3)-Transformer (Appendix 6.6).
> - How is the permutation of tokens handled? Good question! Our method relies on the long-convolutional framework which is not permutation equivariant (except for cyclic permutations) and hence requires canonical ordering. This nuance was elaborated in the Limitations and Future work section (lines 535-539).
>
> [1] Poli et al. Hyena Hierarchy: Towards Larger Convolutional Language Models. 2023. \
> [2] Gu and Dao. Mamba: Linear-Time Sequence Modeling with Selective State Spaces. 2023
>
> ### Reproducibility:
> We appreciate the reviewer’s emphasis on reproducibility. In alignment with our organization's/institution’s policy, we are permitted to release code and data only after the paper’s acceptance.
>
>
> We really appreciate your feedback and comments on improving the clarity and readability of our paper! We are fully committed to keep working towards making the presentation more straightforward and accessible!

---

> ### Author Response · Authors · 2024-11-27
> **Author clarification**
>
> We noticed that our previous response about computational complexity contained a typo that is now fixed. The computational complexity of our method is never quadratic and is O(N log N) due to the FFT convolution.

---

> > ### Author Response · Authors · 2024-11-29
> > **Author response**
> >
> > As the discussion period draws to a close, we would like to check in on whether you've had a chance to review our responses and have any follow up questions?
> >
> > We hope that our reply clarifies and alleviates the reviewer’s concerns. If this is the case, we kindly ask the reviewer to consider raising their score.

---

> > > ### Comment · Reviewer_ZLS9 · 2024-12-02
> > > **Comment on Author Responses**
> > >
> > > Thanks for your response. Please see my comments below.
> > >
> > > > 1. Experiments on small-scale QM9 and MD17 datasets.
> > >
> > > I disagree with the authors' comments. Instead, the experiments on these datasets (even just on one of them) can provide some information. For example, the proposed method can work better on datasets containing large molecules while potentially performing on par with (or slightly better than) other methods on QM9 and MD17. This can demonstrate when the proposed method works.
> > >
> > > > 2. Additional results comparing equivariant Transformers and others.
> > >
> > > I don't think your setting is correct. Those methods (e.g., SE(3)-Transformer and Equiformer) use local context (i.e., cutoff radius) instead of global attention. That is more like hyper-parameter that we can turn on or off depending on problems and compute. Using nearest neighbor graphs and smaller cutoff radius already exist. The main point is what is the strength of the proposed method when it is combined with other previous methods.
> > >
> > > > 3. Type-0 vectors and "one channel" of type-1 vector.
> > >
> > > I disagree with what you say. Using higher degrees is theoretically better, and when trained on large amounts of datasets, it can lead to better empirical results. The issue of longer training/inference time can be addressed by better algorithms (e.g., eSCN) and better implementations (e.g., cuEquivariance) while the lack of expressivity cannot be addressed by other approaches.
> > >
> > > > 4. Cartesian space and type-0 and type-1 vectors.
> > >
> > > Please double check the literature or provide a very concrete example.
> > >
> > >
> > > At this stage, I keep my rating since most of my comments are not addressed.

---

> ### Author Response · Authors · 2024-12-03
> **Author response**
>
> ### Experiments on small-scale QM9 and MD17 datasets
> We want to emphasize that the goal of our work is to tackle large-scale geometric systems that are beyond the reach of existing equivariant models with global context due to computational limitations. The experiments on small-scale datasets such as QM9 or MD17 do not present the same scalability challenges that our model is designed to overcome and thus are not appropriate benchmarks to evaluate our method. Nevertheless, we appreciate your suggestion and recognize that extending experimental comparison will further support the contributions of our work.
>
> ### Type-0 vectors and "one channel" of type-1 vector
> Thank you for your feedback regarding higher-order representations. We completely agree that higher-order features are theoretically more expressive and can lead to better empirical results. Importantly, our method can be generalized to include higher-order features, and we have provided a blueprint for this generalization in Appendix 6.6, utilizing steerable representations or the scalarization trick. We believe that integrating higher-order features into our framework is a promising direction for future work and could further enhance the capabilities of our model.
>
> However, the focus of our work is on computational efficiency and scalability. To the best of our knowledge, no existing method employ higher-order (with l>1) features for truly large-scale real-world datasets due to the corresponding computational complexity. While efficient implementations can mitigate some of these challenges, developing such implementations is highly non-trivial. For example, cuEquivariance, which the reviewer mentions, was only released in mid-November 2024 (after our paper was already under review), despite the widespread use of equivariant networks for many years. This underscores the complexity involved in creating efficient implementations for higher-order representations.
>
> ### Experimental setting comparing equivariant Transformers
> We are not sure we fully understand the comment about the setting. We recognize that methods like SE(3)-Transformer and Equiformer can be configured to use local context by introducing a cutoff radius, thereby limiting their attention to a local neighborhood. However, in our experimental setup, we did not apply any cutoff radius to SE(3)-Transformer or Equiformer, allowing them to attend to the entire context. This was intentional to evaluate their performance as global context models in contrast to the methods that are specifically designed to operate locally (denoted as local and global methods in Table 1).
>
> ### Cartesian space and type-0 and type-1 vectors
> From [1] Equation 4: we can convert any vector $\mathbf{x} \in \mathbb{R}^3$ into type-l vector through the evaluation of spherical harmonics $Y^{(l)}_m : S^2 \rightarrow \mathbb{R} $ at $\mathbf{x}/||\mathbf{x}||$. For any  $\mathbf{x} \in \mathbb{R}^3$:
>
> $\tilde{\mathbf{a}}^{(l)}= \left(Y^{(l)}_m\left(\tfrac{\mathbf{x}}{\lVert \mathbf{x} \rVert}\right)\right)^T_m$ where $m=-l,-l+1,...,l$
>
> [1] J. Brandstetter et al. Geometric and Physical Quantities Improve E(3) Equivariant Message Passing. ICLR 2022.
>
> We thank the reviewer for the valuable feedback and comments, that will strengthen our work in the future!

---

### Official Review · Reviewer_Hpjt · 2024-11-04

**Soundness:** 2
**Presentation:** 2
**Contribution:** 2
**Rating:** 3
**Confidence:** 4

**Summary:**

The paper introduces the SE(3)-Hyena operator to capture global geometric information while preserving equivariant constraints. It aims to address the computational limitations of existing methods, such as self-attention and local processing techniques. The proposed model is evaluated on dynamical system modeling and RNA property prediction tasks, and the authors introduce an "equivariant associative recall" task to assess contextual learning abilities.

**Strengths:**

Applying deep learning to problems that involve modeling geometric context, as done in this paper, is a valuable direction in the field. Performance improvements in this area often depend on architectural advances, which is also good to see.

**Weaknesses:**

- The experiments in the paper are not comprehensive enough to clearly demonstrate the advantages of the proposed method over existing ones. For example, baselines used in the paper -- SchNet, EGNN, and SE(3)-Transformer -- have been evaluated on the QM9 dataset in their original papers. It would be more convincing if the authors included results on QM9 as well.

- Several state-of-the-art baselines for dynamical system modeling are missing, making the performance of the proposed model not convincing enough. Examples include SEGNN [1], SAKE [2], SEGNO [3], and GeoMFormer [4].

[1] Geometric and Physical Quantities Improve E(3) Equivariant Message Passing. ICLR 2022.

[2] Spatial Attention Kinetic Networks with E(n)-Equivariance. ICLR 2023.

[3] Improving Generalization in Equivariant Graph Neural Networks with Physical Inductive Biases. ICLR 2024.

[4] GeoMFormer: A General Architecture for Geometric Molecular Representation Learning. ICML 2024.

**Questions:**

Since the work focuses on "scalable" equivariant learning, could the authors provide results on larger-scale datasets to further demonstrate the model's effectiveness?

---

> ### Author Response · Authors · 2024-11-25
> **Author response**
>
> We appreciate the reviewer's recognition of the value of architectural advances in the field of modeling geometric context. Next, we would like to address the weaknesses raised by the reviewer:
>
> ###  Extending experimental comparison.
> We absolutely agree with the reviewer that the experimental comparison will benefit from more baselines. We have now included results for five additional baseline models: LEFTNet, FastEGNN, TorchMD-NET Equivariant Transformer, Equiformer, and Vector Neuron Transformer (VNT) to the two RNA property prediction datasets. The updated results are presented in Table 1 of the revised paper. Our SE(3)-Hyena model still considerably outperforms both local and global context baselines, which we attribute to its mechanism of alternating between local and global context (quantitatively supported in Table 3, Appendix 6.4). Moreover, while global context baselines like Equiformer, VNT, and SE(3)-Transformer encounter memory limitations with all-atom representations, SE(3)-Hyena still can process large-scale all-atom representation, further improving prediction quality on the Open Vaccine dataset. We additionally include the runtime and memory comparison with new global context baselines (Figure 6) in the revised version. These updated results support our conclusion that SE(3)-Hyena effectively models global geometric context at scale.
>
> Also, we now include other global context baselines (Vector Neuron Transformer [6] and Equiformer [7]) in the associative recall experiment. These baselines perform on par with SE(3)-Hyena in the fixed vocabulary setting but are significantly worse in the harder random vocabulary case where a model must learn an equivariant retrieval mechanism. We attribute the superior performance of the SE(3)-Hyena in this task to the alternating usage of both local and global context, so the model can first associate key-value bi-grams locally, and then compare those bi-grams on a whole sequence level. It demonstrates that not all equivariant models can solve the associative recall equally well. The updated results are in the revised manuscript.
>
>
> ### Experiments on QM9 dataset and additional baselines on N-body
> We wish to clarify that the focus of our work is on large-scale geometric systems that require modeling global context with computational efficiency. For instance, the associative recall experiment involves geometric systems with up to 2^10 tokens, while the RNA experiments require modeling systems containing 5800 to 11300 atoms. In contrast, the QM9 dataset consists of small molecules with up to 30 atoms, and the N-body problem contains only 5 moving particles in its standard version. These small-dimensional datasets present fundamentally different challenges and requirements to the model than what is required for global context modeling of large geometric systems. Consequently, methods optimized for QM9 or N-body (such as Equiformer) face computational limitations (Figure 6 in the revised text) which renders them inapplicable for large-scale datasets. With this, we do not consider the QM9 or N-body as representative tasks to comprehensively evaluate the capabilities of our model, and benchmark it against other methods.
>
> Nevertheless, we still chose to provide the N-body experiment (now moved to Appendix 6.2) as it is widely used in existing literature as a toy starting problem to showcase the capabilities of equivariant architectures, and we appreciate the reviewer's suggestion for more baselines which we now include into experimental comparison.
>
> ### Q1: Could the authors provide results on larger-scale datasets to further demonstrate the model's effectiveness?
> We wish to highlight that that the RNA dataset used in the paper contains mRNA sequences of from 5800 up to 11300 atoms which is considered large-scale. We clarified this in the description of the datasets in the paper. We would also appreciate if the reviewer can point us to other large-molecule datasets with property labels.

---

> > ### Author Response · Authors · 2024-11-29
> > **Author response**
> >
> > As the discussion period draws to a close, we would like to check in on whether you've had a chance to review our responses and have any follow up questions?
> >
> > We hope that our reply clarifies and alleviates the reviewer’s concerns. If this is the case, we kindly ask the reviewer to consider raising their score.

---

> > ### Comment · Reviewer_Hpjt · 2024-12-02
> >
> > Thank you for the response. The experiments in the paper remain quite limited. In the revised version, the authors removed the N-body simulation (dynamical system modeling) experiments from the main text after reviewers highlighted the lack of many recent baselines that outperform the proposed model. The only remaining real-world experiment is RNA property prediction. However, both the limited number of experiments and the lack of diversity in the data domain make it insufficient to justify the paper's contributions, so I will keep my score.

---

> > > ### Author Response · Authors · 2024-12-02
> > > **Authors response**
> > >
> > > We thank the reviewer for the response. We want to highlight that all the recent baselines asked by reviewers are included in the N-body experiment. The N-body experiment was moved to the Appendix because (i) our work tackles large-scale geometric systems, and standard 5-particle N-body experiment is not a representative task for modeling large-scale geometric systems; (ii) to accommodate the new results and reviewers' suggestion within 10 pages limit.
> > >
> > > Also, we want to highlight that associative recall is a widely recognized experiment, routinely used to evaluate the contextual learning capabilities of sequence models. For instance, it has been used to benchmark models such as Hyena [1] and even serves as the primary motivation for models like Mamba (Figure 2 in [2]). Furthermore, recent studies demonstrate that mechanistic interpretability tasks are indicative of a model's scaling behavior [3], where improved performance on tasks such as associative recall directly correlates with better generalization to real-life downstream tasks. Given its broad acceptance in the literature, we consider associative recall a serious experiment to evaluate models' context learning abilities.
> > >
> > > [1] Poli et al. Hyena Hierarchy: Towards Larger Convolutional Language Models. 2023. \
> > > [2] Gu and Dao. Mamba: Linear-Time Sequence Modeling with Selective State Spaces. 2023 \
> > > [3] Poli et al. Mechanistic Design and Scaling of Hybrid Architectures. 2024.
> > >
> > > We also kindly ask the reviewer to consider **all dimensions of our work, including theoretical and methodological contributions**, alongside the experimental part, which is important but not the only component of the paper.

---

> > > > ### Comment · Reviewer_Hpjt · 2024-12-03
> > > >
> > > > I have reviewed the author's response.
> > > >
> > > > Theoretical and methodological "contributions" hold value only when supported by experimental evaluation on diverse real-world datasets. Without such evidence, the theoretical and methodological parts of a work add little to the community.
> > > >
> > > > Similarly, associative recall is a synthetic task and it cannot replace the necessity of diverse real-world evaluations. Its results are meaningful only when complemented by performance on diverse real-world datasets.
> > > >
> > > > The limited number and lack of diversity in the real-world experiments remain to be my concerns, and the author's response does not adequately address this critical issue. Therefore, I maintain my rating.

---

> > > > > ### Author Response · Authors · 2024-12-03
> > > > > **Author response**
> > > > >
> > > > > We appreciate the reviewer's emphasis on experimental validation of the method. We would like to highlight that our RNA property prediction experiments involve two large-scale, real-world datasets that presents significant challenges for existing methods that our model is designed to address. We believe this experiment showcases the applicability and effectiveness of our method on complex, real-world data. Regarding the associative recall task, we understand that synthetic tasks cannot replace evaluations on diverse real-world datasets. Our intention was to use this task as a complementary benchmark to assess the contextual learning capabilities of our model, and not as a substitute for real-world experiments.
> > > > >
> > > > > With this, we understand the reviewer's concern about extending experimental comparison. We agree that including a broader range of datasets would further strengthen our work, and we are committed to extending our experimental evaluations in future research.
> > > > >
> > > > > Thank you for your feedback and for the time you've invested in reviewing our work!

---

### Official Review · Reviewer_WsDB · 2024-11-06

**Soundness:** 3
**Presentation:** 2
**Contribution:** 2
**Rating:** 6
**Confidence:** 3

**Summary:**

The paper introduces the SE(3)-Hyena operator, the first equivariant long-convolutional model with sub-quadratic complexity for global geometric context.  Importantly, authors claim their framework is flexible and can accommodate any equivariant network as the projection function.

**Strengths:**

The idea of using global information to improve the model is very natural, and the convolution simplification of the cross product is very elegant.

**Weaknesses:**

> **W1. Lack of discussion on related work.**

There are many works that use global features to improve equivariance. Although this paper's work is obviously different from them, it is recommended to add a discussion on these works (e.g. FastEGNN [a], Neural P^3M [b]).

> **W2. The motivation for operator design is unclear.**

Why is the motivation for using the cross product not well explained? As we all know, the cross product is the Hodge star dual of the outer product. Can it be explained from this perspective? In addition, the introduction of the cross product actually produces pseudovectors, which also leads to the fact that this paper is only SE(3) equivariant rather than E(3) equivariant. Is such an introduction really reasonable?

> **W3. Results on N-body lack the latest baseline.**

Some of the latest results are not shown (e.g. 0.0043 of SEGNN [c], 0.0039 of CGENN [d]). Compared with these works, the results of this work seem to be insufficient.

> **W4. The significance of associative recall experiments is unclear.**

The current experiment cannot illustrate the "contextual learning capabilities of sequence models" that the authors want to claim. First, the model lacks more baselines (e.g. LEFTNet [e], MACE [f], EquiformerV2 [g], SO3krates [h]). It is more like explaining that in this setting, equivariant models are better than models without built-in symmetry priors. Secondly, this experiment lacks practical application significance. I can't seem to find a corresponding task in real life. I hope the authors can give further explanation.

> **W5. Lack of baseline in RNA dataset.**

The baseline is also missing, and the baseline mentioned in W1 and W4 should be supplemented.

> **W6. Lack of expansion on other models.**

Authors claim their framework is flexible and can accommodate any equivariant network as the projection function. Is it possible to extend several common models (e.g. SchNet [i], EGNN [j], MACE [f], HEGNN [k])?

> **W7. Others (Some typos)**

- Line 144: where is the function $\Psi$, or the hat $\hat{\mathbf{x}_i}, \hat{\mathbf{f}_i}$ are the outputs?
- Line 231: in calculation of $\alpha_3$, it should be $\mathbf{r}_1^\top\mathbf{r}_2$
- Line 242: feature tuples $f_i$, LaTeX misses underscore
- Line 817: "hiddent dimension" should be "hidden dimension"
- Others: the logarithmic symbol should be $\log$ instead of $log$, and why is there a base sometimes without and sometimes with 2?

[a] Improving Equivariant Graph Neural Networks on Large Geometric Graphs via Virtual Nodes Learning

[b] Neural P3M: A Long-Range Interaction Modeling Enhancer for Geometric GNNs

[c] Geometric and Physical Quantities Improve E(3) Equivariant Message Passing

[d] Clifford Group Equivariant Neural Networks

[e] A new perspective on building efficient and expressive 3D equivariant graph neural networks

[f] Mace: Higher order equivariant message passing neural networks for fast and accurate force fields

[g] EquiformerV2: Improved Equivariant Transformer for Scaling to Higher-Degree Representations

[h] A Euclidean transformer for fast and stable machine learned force fields

[i] SchNet: A continuous-filter convolutional neural network for modeling quantum interactions

[j] E(n) Equivariant Graph Neural Networks

[k] Are High-Degree Representations Really Unnecessary in Equivariant Graph Neural Networks?

**Questions:**

See Weakness.

---

> ### Author Response · Authors · 2024-11-25
> **Author response (1/2)**
>
> We appreciate the reviewer highlighting the elegance of the proposed convolution simplification of the cross product as this is one of the core contributions of our work. Next, we would like to address the weaknesses raised by the reviewer:
>
> ### W1: Lack of discussion on related work.
> We appreciate the reviewer's suggestion to extend the related work with the relevant methods. We have now additionally elaborated on the existing works that rely on global features to improve equivariance, including recent FastEGNN and Neural P^3M.
>
> ###  W2a. The motivation for operator design is unclear.
> The core operation for the SE(3)-Hyena is the vector long-convolution where the cross product is used to combine two vectors (one from kernel and one from signal) into a third vector of the resulting vector-valued signal. Alternatively, we can consider a more general notion of the Wedge product which will combine two vectors in $\mathbb{R}^3$ into a bivector. Then, to map a bivector back to $\mathbb{R}^3$, we need to take its Hodge star dual. It is known that the Hodge star dual of the wedge product between two vectors precisely reassembles the cross product between them, motivating the usage of cross product in this work. We appreciate the reviewer’s observation that this provides a broader intuition for the choice of cross product, and we are happy to elaborate further on this in the paper.
>
> ###  W2b. SE(3) rather than E(3).
> The reviewer is absolutely right that the cross product produces pseudovectors, leading to the SE(3) rather than E(3). We want to clarify that this is intentional, as the additional equivariance or invariance to reflections, introduced by E(3), is a key limitation when processing chemical or biological systems where the chirality is a concern as a molecule and its reflection (enantiomer or diastereomer) can come with drastically different properties and characteristics.
>
> ###  W3. Results on N-body lack the latest baseline.
> We appreciate the reviewer's suggestion for SEGNN and CGENN N-body baselines, and we have now included them in the N-body comparison.
>
> However, we wish to clarify that the focus of our work is on large-scale geometric systems that require modeling global context with computational efficiency. Thus, we do not consider the N-body problem as the representative task to comprehensively evaluate the capabilities of our model, and benchmark it against other methods. While the N-body task is useful to demonstrate to what extent one's model can be optimized in a synthetic well-controlled environment, it says little about the model's capabilities to handle long-range context at scale. Nevertheless, we still chose to provide the N-body experiment (now moved to Appendix 6.2) as it is widely used in existing literature as a toy starting problem to showcase the capabilities of equivariant architectures.
>
> ###  W4. The significance of associative recall experiments is unclear.
> We want to highlight that associative recall is a widely recognized task within the mechanistic interpretability suite [1], routinely used to evaluate the contextual learning capabilities of sequence models. For instance, it has been used to benchmark models such as Hyena [2] and even serves as the primary motivation for models like Mamba [3] (Figure 2). Furthermore, recent studies demonstrate that mechanistic interpretability tasks are indicative of a model's scaling behavior [4], where improved performance on tasks such as associative recall directly correlates with better generalization to real-life downstream tasks. Given its broad acceptance in the literature, we consider associative recall a meaningful measure of the model's context learning abilities.
>
> Following the reviewer's suggestion, we include other global context baselines (Vector Neuron Transformer and Equiformer) in the associative recall experiment. These baselines perform on par with SE(3)-Hyena in the fixed vocabulary setting but are significantly worse in the harder random vocabulary case where a model must learn an equivariant retrieval mechanism. We attribute the superior performance of the SE(3)-Hyena in this task to the alternating usage of both local and global context, so the model can first associate key-value bi-grams locally, and then compare those bi-grams on a whole sequence level. It demonstrates that not all equivariant models can solve the associative recall equally well. The updated results are in the revised manuscript.
>
> [1] Olsson et al. In-context learning and induction heads. 2022.\
> [2] Poli et al. Hyena Hierarchy: Towards Larger Convolutional Language Models. 2023.\
> [3] Gu and Dao. Mamba: Linear-Time Sequence Modeling with Selective State Spaces. 2023\
> [4] Poli et al. Mechanistic Design and Scaling of Hybrid Architectures. 2024.

---

> ### Author Response · Authors · 2024-11-25
> **Author response (2/2)**
>
> ###  W5. Lack of baseline in RNA dataset.
> We agree with the reviewer that the RNA property prediction experiment would benefit from additional baselines. We have now included results for five additional models: LEFTNet, FastEGNN, TorchMD-NET Equivariant Transformer, Equiformer, and Vector Neuron Transformer (VNT). The updated results are presented in Table 1 of the revised paper. Our SE(3)-Hyena model considerably outperforms both local and global context baselines, which we attribute to its mechanism of alternating between local and global context (quantitatively supported in Table 3, Appendix 6.4). Moreover, while global context baselines like Equiformer, VNT, and SE(3)-Transformer encounter memory limitations with all-atom representations, SE(3)-Hyena still can process large-scale all-atom representation, further improving prediction quality on the Open Vaccine dataset. We additionally include the runtime and memory comparison with new global context baselines (Figure 6) in the revised version. These updated results support our conclusion that SE(3)-Hyena effectively models global geometric context at scale.
>
> ###  W6. Lack of expansion on other models:
> Our framework indeed can accommodate any equivariant network as the projection function. Note, that our method is already using EGNN (slightly modified to accommodate global messages as explained in Section 3) to equivariantly project input tokens. The choice of EGNN is motivated by its balanced performance and high computational efficiency compared to higher-order methods such as MACE or HEGNN. We acknowledge that incorporating more advanced equivariant networks as the projection layer could further enhance performance. Due to time constraints in this rebuttal, we highlight this as a direction for future work in the revised version.
>
> ###  W7. Typos and clarifications:
> We thank the reviewers for these suggestions! Line 144: \Psi is the SE(3)-Hyena operator defined right above the formula. Line 231: the reviewer is absolutely right $\alpha_3$ indeed accepts $r_1^T r_2$, we fixed the typo. We fixed typos in lines 242 and 817, and we fixed the log symbol. We use the standard notation of $\log$ to denote the natural logarithm, and $\log_2$ to denote the logarithm with base 2. The logarithm with base 2 is needed to describe the computational complexity of FFT convolution.

---

> ### Comment · Reviewer_WsDB · 2024-11-25
>
> The authors' responses resolved most of my questions, and I will raise my score to "weakly accept". However, there are still some less important residual questions that would like to be answered (mainly D3).
>
> > **D1. Explanation of experimental results**
>
> I saw that the author compared FastEGNN, and it seems that this way of introducing virtual nodes did not bring about performance improvement. Can the author discuss the reason for this?
>
> From my personal understanding, the virtual nodes introduced by FastEGNN do not contain priors, so they are not suitable for data such as RNA that naturally have sequence priors, and may require specialized design. I suggest authors to further explain the superiority of its SE(3)-Hyena design from this perspective (for example, the long convolution design can make it well adapted to sequence data).
>
> I think this presentation is a respect for the original authors of FastEGNN on the one hand, and on the other hand, it can also improve readers' understanding of the advantages of SE(3)-Hyena.
>
> In addition, I also noticed that some of the ICLR'25 submissions used hierarchical methods to model RNA, such as EquiRNA [a]. I wonder if SE(3)-Hyena can also implicitly learn some hierarchical knowledge?
>
> [a] Size-Generalizable RNA Structure Evaluation by Exploring Hierarchical Geometries
>
> > **D2. The  recall experiments**
>
> I am very grateful to the authors for their answers to W4. I really don't know much about this task. I appreciate their explanations, although I still feel a little confused, but I believe it is not the authors' fault. I will leave this aside and re-evaluate the article.
>
> > **D3. Expansion on other models**
>
> Since the experiments added by the authors are indeed very solid (which means time-consuming), I understand that the authors do not have enough time to further discuss how to expand. In fact, what I want to know is that the core operation of SE(3)-Hyena is aimed at pseudovectors. How to expand it on high-degree steerable features?
>
> Let I take HEGNN as an example. It is very simple. HEGNN just adds the inner product of high-degree steerable features to the scalar of EGNN message calculation (thereby avoiding tensor product) and keeps the update formula of linear combination. What I want to know is whether and how the pseudovectors generated by SE(3)-Hyena will interact with these high-degree steerable features. Can the authors have a simple **theoretical discussion** (no experiments are required)?
>
> > **DO. Others**
>
> - Line. 366: should " eun" be "run"?
> - The complexity seems to have negligible constant effects, so could the authors drop the logarithmic base to make it look simpler?

---

> ### Author Response · Authors · 2024-11-26
> **Author response**
>
> ### D1. Explanation of experimental results
>
> - Virtual nodes mechanism and advantages of SE(3)-Hyena.
>
> As the difference between EGNN and FastEGNN is in the usage of virtual nodes, and since the virtual nodes are observed useful in other tasks (such as those in FastEGNN paper), indeed we can assume that this virtual node mechanism may not be optimal for RNA molecules which, as reviewer notices, come with a strong sequence prior. In particular, Center of Mass (CoM) initialization can be sub-optimal for large molecules such as RNA due to intricate geometric structures with multiple segments extending far from the CoM (as illustrated in Figure 1). This may impede the stability of FastEGNN training due to its distance-based update rule. Addressing this issue may require a more specialized design, as the reviewer suggests. In contrast, SE(3)-Hyena is directly related to long-convolutional and state-space models, which naturally are sequence models. A unique advantage of long convolution is its ability to discriminate between different orderings of sequence tokens by breaking permutation symmetry — a feature particularly well-suited for molecules with a strong sequence prior. We thank the reviewer for bringing this point forward, and we now have elaborated on this in the manuscript (Section 4.2 Results).
>
> - Hierarchical knowledge.
>
> This is a great question! The SE(3)-Hyena indeed should be able to learn hierarchical knowledge to solve the RNA tasks presented by Open Vaccine and Ribonanza datasets which require modeling properties which, among others, depend on the connections between nucleotides (secondary structure). The strong performance of our model indicates its ability to implicitly learn the hierarchical nature of RNA up to the nucleotide level from atom level. We have additionally elaborated on the hierarchical aspect of the RNA data in Appendix 6.3.3. Also, thank you for the reference! [a] presents a very interesting approach to RNA and will indeed be a good citation to emphasize the challenge of hierarchy modeling in RNA molecules. We have now included it in Appendix 6.3.3.
>
> ### D3. Expansion to higher-order steerable features
>
> Good question! This is a very interesting direction for future research. Extending the SE(3)-Hyena to higher-order steerable features requires working out steerable vector long convolution. Conceptually, the same mathematical approach can be applied but instead of vectors in $\mathbb{R}^3$ and cross product, we will have steerable vectors and tensor product. Then, the long convolution can be built with tensor product and reduced to a series of FFT scalar convolutions by factoring out CG coefficients. This is the overall blueprint of one of the possible approaches. In this approach, various types of steerable vectors will interact via standard or parametrized tensor product.
>
> Alternatively, it is also possible to use the scalarization trick to model high and low order feature interaction. This can be done as a direct extension of the geometric long convolutional framework with the introduction of a higher-order part (i.e. we make $(a,x)$ feature tuple into $(a,x,v)$ where a - scalar, x - vector in $\mathbb{R}^3$, v - higher-order vector parts), and dropping higher-order to higher-order interaction term in the product between two feature tuples.
>
> ### DO. Others
> We fixed the typo in Line. 366. The logarithmic base indeed can be dropped for the sake of simple notation, since the computational complexity of FFT convolution is well-known. Thanks for your suggestions!

---

> ### Comment · Reviewer_WsDB · 2024-11-27
>
> Thank you for your response. I am satisfied with the authors' explanation of D3, which has addressed my previous questions. However, since there isn't a weak accept choice (7) in ICLR, I will consider the feedback from other reviewers and your reply to decide whether to further improve the rating.
>
> I suggest that the authors list this blueprint in the appendix of this article and explain the two types of models separately. This will give later readers inspiration and thinking, and also declare the first discovery of this technique.
>
> As the author said, for models using CG tensor products (e.g. TFN, SEGNN and MACE), SE(3)-Hyena can consider building a global steerable context in a similar way, while for high-order models using scalarization techniques (e.g. HEGNN and GotenNet [a]), SE(3)-Hyena can first build a global scalar context and then build a global steerable context.
>
> In fact, techniques like HEGNN and GotenNet that use scalarization techniques to introduce high-order features may be a very important research direction in the future, because their complexity itself is only $\mathcal{O}(L^2)$, where $L$ represents the highest degree of high-degree steerable features. It is worth noting that GotenNet submitted to ICLR25 has achieved SOTA results on small molecules such as QM9. I believe that if it can be combined with the concise but efficient global context representation operator such as SE(3)-Hyena, it will be very helpful to bring this success to the task of large-scale geometric graphs.
>
> [a] Rethinking Efficient 3D Equivariant Graph Neural Networks

---

> > ### Author Response · Authors · 2024-11-28
> > **Author response**
> >
> > We are happy that our response addressed all reviewer's questions and concerns.
> > We totally agree that combining scalarized higher-order representations with efficient global context representation operators such as SE(3)-Hyena presents a promising direction for future research! We have now listed the blueprint in Appendix 6.6 in the revised manuscript and organized it into two subsections: higher-order steerable representations and faster higher-order representations by scalarization.
> >
> > We thank the reviewer for continuing engagement in the discussion, and openness to further raise the score higher than weak accept.

---

### Note · Authors · 2024-12-06

I have read and agree with the venue's withdrawal policy on behalf of myself and my co-authors.